# Dysfunctional mitochondria in ageing T cells: a perspective on mitochondrial quality control mechanisms

Lin Luo [ID] [1,2,3], Ana Victoria Lechuga-Vieco [ID] [4], Clara Sattentau[5], Mariana Borsa[3] & Anna Katharina Simon [ID] [1,3 ✉]

## Abstract

**Dysfunctional mitochondria are a hallmark of T cell ageing and contribute to organismal ageing. This arises from the accumulation of reactive oxygen species (ROS), impaired mitochondrial dynamics, and inefficient removal of dysfunctional mitochondria. Both cell-intrinsic and cell-extrinsic mechanisms for removing mitochondria and their byproducts have been identified in T cells. In this review, we explore how T cells manage mitochondrial damage through changes in mitochondrial metabolism, mitophagy, asymmetric mitochondrial inheritance, and mitochondrial transfer, highlighting the impact of these mechanisms on T cell ageing and overall organismal ageing. We also discuss current therapeutic strategies aimed at removing dysfunctional mitochondria and their byproducts and propose potential new therapeutic targets that may reverse immune ageing or organismal ageing.**

**Keywords** T Cell Ageing; Mitochondrial Metabolism; Mitophagy; Asymmetric Cell Division; Mitochondrial Transfer
**Subject Categories** Immunology; Molecular Biology of Disease; Organelles

## Introduction

The ageing population is increasing at an unprecedented rate, with the population of over 60-year-olds projected to reach 2.3 billion by 2050 (WHO). Ageing is a significant risk factor for multiple morbidities, including diabetes, cardiovascular conditions, neurological deficits, autoimmune diseases and susceptibility to infection and cancer (Ellison-Hughes, 2020). This has given rise to research into an array of therapeutics targeting the fundamental ageing process. Ageing affects all biological systems, and the impaired function of the immune system has been recognised as one of its principal features (Yousefzadeh et al, 2021). Emerging evidence suggests that an aged immune system contributes to cellular senescence and the ageing of solid organs (Yousefzadeh et al, 2021; López-Otín et al, 2023). Dysfunctional cell and tissue metabolism, in particular, has been identified as a key factor expanding the hallmark of both immune decline and ageing (López-Otín et al, 2023). Despite ongoing research, our understanding of the aged immune system, the mechanisms driving its dysfunction, and its role in age-related diseases remains incomplete.

During ageing, the immune system becomes less effective at combating infections and cancer. Age-related alterations in the immune system are usually characterised by a decline of function among immune cells and changes including lower naïve/memory T cell ratio, thymus atrophy resulting in less naïve T cell output, and decreased avidity of antibody response (Pawelec and Solana, 1997; Chen et al, 2022; Walford, 1964). Aged immune cells share a common senescent phenotype encompassing increased cyclin-dependent kinase inhibitor (CDKN) expression, including p16 and p21 and senescence-associated secretory phenotype (SASP) that includes the release of inflammatory cytokines and chemokines (Callender et al, 2018). A further attribute of an aged immune system is defined by chronic and systemic inflammation, known as inflammageing (Jin et al, 2023). Counterintuitively, inflammageing challenges the limited capacity of an aged immune system to resolve infection. For example, cytomegalovirus (CMV) infection, typically asymptomatic, becomes symptomatic in older women due to high levels of inflammatory cytokines that activate the replication of cytomegalovirus (Schmaltz et al, 2005). The release of pro-inflammatory cytokines also supports tumour growth and metastasis, resulting in higher susceptibility to cancers (Tato-Costa et al, 2016; Coppé et al, 2008). These features are used to define the interdisciplinary concept of immune senescence, which perpetuates cellular and environmental ageing, ultimately leading to organismal decline.

Amongst the array of cells in the immune system, T lymphocytes generate specific and diverse immune responses towards antigens, playing a central role in defending the body against both infections and cancer. Together with B lymphocytes, they form immunological memory following encounter with an antigen to arm the body for re-exposure to the same antigen. A

[1]Max-Delbrück-Center for Molecular Medicine in the Helmholtz Association (MDC), Berlin, Germany. [2]CAMS Oxford Institute, Nuffield Department of Medicine, University of Oxford, Oxford, UK. [3]Kennedy Institute of Rheumatology, Nuffield Department of Orthopaedics, Rheumatology, and Musculoskeletal Sciences, University of Oxford, Oxford, UK. [4]Institute for Research in Biomedicine (IRB Barcelona), The Barcelona Institute of Science and Technology, Barcelona, Spain. [5]East Cheshire NHS Trust, Macclesfield, Cheshire East, UK. ✉E-mail: katja.simon@mdc-berlin.de

decline in immunological memory formation has been acknowledged as a key change during immune ageing (Haynes et al, 2003). Both cytotoxic T cell (CD8+) and T helper cell subsets (CD4+) emerge from the thymus as naïve cells. Cytotoxic CD8+ T cells recognise antigens presented by major histocompatibility complex (MHC) molecules I on the surface of infected or cancerous cells, which are then targeted for killing through effector molecules. CD4+ T helper cells recognise antigen through MHC class II and support the activation and persistence of CD8+ T cells, enhance antigen presentation by dendritic cells, and recruit other immune effector cells to the infection site or the tumour microenvironment. This coordinated response is also a fundamental component of the immune system's cancer surveillance mechanisms, although tumour antigens are more difficult to target due to their immune evasion and suppression potentials (Sun et al, 2023; Cornel et al, 2020). After T cells' first encounter with antigen, they become effector cells and memory precursors, the latter giving rise to long-lived memory cells (Sun et al, 2023).

Existing evidence substantiates the role of metabolic adaptation during T cell activation, survival, and differentiation to fuel their functional demands (Raynor and Chi, 2024; Chapman et al, 2020). Naïve T cells rely on oxidative phosphorylation (OXPHOS) to maintain their quiescence with efficient ATP production. Effector cells rely primarily on aerobic glycolysis for rapid metabolic mobilisation, while memory T cells rely on fatty acid oxidation (FAO) to reserve their metabolic capacity (Gubser et al, 2013; Corrado and Pearce, 2022; O'Sullivan et al, 2014).

Metabolic imbalance is a key driver contributing to the anergy of T cell function during ageing (Zheng et al, 2009; Powell and Delgoffe, 2010; Desdín-Micó et al, 2020; Callender et al, 2020). These metabolic changes are closely related to mitochondrial dysfunction, which disrupts their multifaceted role in maintaining T cell homeostasis. Cell homeostasis relies on the tight quality control of energy sensing/production and metabolism regulation by mitochondria. Mitochondria are highly dynamic organelles that serve not only as the primary producers of cellular energy, but also as metabolic hubs that rewire nutrient sensing and metabolic pathways, including aerobic glycolysis, glutaminolysis, pentose-phosphate pathway (PPP) activity, and one-carbon metabolism (Han et al, 2023). They generate the key metabolites to build functional molecules (proteins, membranes and nucleic acids) and energy (adenosine triphosphate (ATP)), which is coupled with reactive oxygen species (ROS)($O_2^-$ and $H_2O_2$) production. In many cell types or model organisms, accumulation of dysfunctional mitochondria induces an ageing phenotype (Wang et al, 2021; Desdín-Micó et al, 2020; Callender et al, 2020; Picca et al, 2023). In T cells, an important consequence of the age-associated mitochondrial dysfunction is the metabolic switch from the tricarboxylic acid (TCA) cycle to glycolysis, which affects their stemness and effector function. Notably, CD8+ T cells show greater age-related decline than CD4+ T cells, with significant depletion of the naïve compartment due to severe mitochondrial dysfunction. These defects lead to reduced metabolic flexibility, impaired electron transport, and elevated ROS, propagating oxidative stress across cellular compartments. The elevated ROS production and oxidative stress by damaged mitochondria hinder T cell differentiation and activation (Wertheimer et al, 2014; Czesnikiewicz-Guzik et al, 2008). A multimorbidity phenotype is observed after genetic knockout of mitochondrial transcription factor A (Tfam) in T cells with reduced life expectancy, systemic dysfunction and immune incompetency in

mice (Desdín-Micó et al, 2020), highlighting that impaired function of T cells alone could predispose older organism to infections, cancer progression or accumulation of senescent cells (Desdín-Micó et al, 2020; Escrig-Larena et al, 2023) (Fig. 1).

Collectively, this suggests a mitochondria-centred role of T cell ageing with an impact on systemic ageing. In this review, we will present how ageing T cells cope with mitochondrial damage by (1) responding to aberrant mitochondrial metabolism and ROS production, (2) mitochondrial degradation, (3) asymmetric mitochondrial inheritance and (4) mitochondrial transfer between cells. Where it's known, we will discuss how these partly still controversial cellular processes impact T cell ageing, hoping to provide new perspectives for anti-ageing or rejuvenation therapies.

# Mitochondrial metabolism during T cell ageing

Age-related changes in T cell phenotypes are closely linked to dysregulation of mitochondrial function. These mitochondrial alterations result in reduced inter-organelle communication,

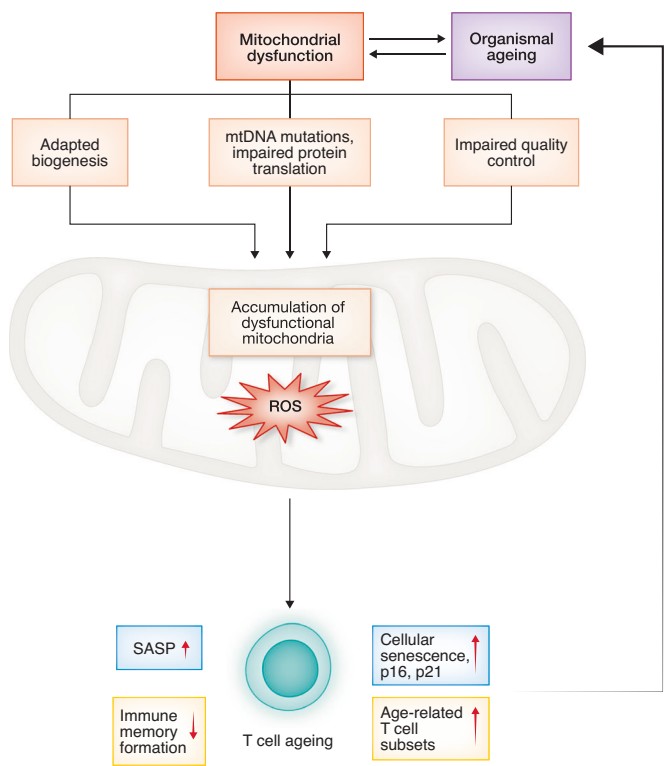

**Figure 1. Mitochondrial failure as a central driver of T cell ageing.**

Ageing impairs mitochondrial function by disrupting biogenesis, genome integrity, protein translation, and quality control mechanisms. As a result, dysfunctional mitochondria accumulate, accelerating both T cell decline of function and organismal ageing. Age-related T cell dysfunction is characterised by diminished immune responses and increased systemic senescence, contributing to inflammageing. T cell senescence is marked by the expression of key regulators such as p16 and p21, as well as the senescence-associated secretory phenotype (SASP), a pro-inflammatory profile that can further impair immune function.

impaired organelle turnover, and a loss of metabolic plasticity and mitochondrial dynamics. In a recent review, the molecular mechanisms underlying dysfunctional mitochondria during T cell ageing have been summarised, including alterations in mitochondrial DNA (mtDNA) genome stability, mitochondrial dynamics, $Ca^{2+}$ homeostasis, mitochondrial biogenesis and degradation and oxidative stress (Escrig-Larena et al, 2023). Here, we will first discuss the role of mitochondrial ROS and mitochondrial biogenesis during T cell ageing.

## Mitochondrial ROS and biogenesis signalling

The accumulation of dysfunctional mitochondria is acknowledged as one of the hallmarks of ageing (López-Otín et al, 2023). ROS production is typically regarded as a byproduct of mitochondrial respiration during ATP production. Both mitochondrial ATP production and ROS generation are closely tied to the availability of oxygen, which is the final electron acceptor within the mitochondrial electron transport chain (ETC). This relies on mitochondrial complexes (I-IV) that help transfer electrons and pump protons across the mitochondrial membrane, creating an energy gradient. The energy gradient is used to synthesize ATP from ADP (adenosine diphosphate) by ATP synthase. During ETC, ROS escapes from the inner mitochondrial membrane, resulting in downstream signalling events and mitochondrial damage (Quinlan et al, 2013), which may trigger permeabilisation of the mitochondrial membrane and oxidative damage, causing inflammaton and cell death (Amorim et al, 2022). However, while ROS-driven signalling has been studied as a pathological factor, ROS as a signalling molecule is as essential as ATP production (Palma et al, 2024).

Besides excessive oxidative stress, alterations in mitochondrial translation and a decline in mitochondrial biogenesis are observed during ageing (Gill et al, 2019; Souder et al, 2023). Maintaining a precise balance between mitochondrial translation and mitochondrial proteostasis is essential for optimal OXPHOS function (Uoselis et al, 2023; Soto et al, 2022). Peroxisome proliferator-activated receptor gamma (PPAR) coactivator-1 alpha (PGC-1α) is a master transcription factor that controls mitochondrial biogenesis. Under conditions of energy deprivation, energy and nutrient-sensing signals such as AMP-activated protein kinase (AMPK) and Sirtuin 1 (SIRT1) are activated, which both directly trigger PGC-1α function by its phosphorylation and deacetylation. When PGC-1α is activated, TFAM helps transporting both SIRT1 and PGC-1α into the mitochondria to regulate mtDNA replication and transcription. SIRT1 is a NAD+ dependent protein deacetylase. As NAD+ levels decline during ageing, the resulting loss of NAD+ impairs SIRT1 activity, which in turn reduces the deacetylation of key regulators such as AMPK, forkhead box O (FOXO) proteins, and PGC-1α. This disruption leads to dysregulated mitochondrial biogenesis and metabolism, as well as decreased antioxidant activity of mitochondrial superoxide dismutase (SOD2), which converts superoxide ($O_2^-$) into hydrogen peroxide ($H_2O_2$). This dysfunction ultimately contributes to increased mitochondrial ROS production (Salminen et al, 2013; Xu et al, 2020). Thus, by lacking SIRT1 activity, ageing results in a scenario where AMPK is activated due to energy-scarce conditions, but this results in mitotic arrest through phosphorylation of p53 and increasing p16 expression (Wiley et al, 2016). As one of the major features of cellular

senescence, mitotic arrest indirectly helps enhance mitochondrial biogenesis: When cells are arrested in mitosis, they shift focus from cell division to repairing damaged organelles, helping to restore mitochondrial function. However, extended mitotic arrest can lead to a decline in mitochondrial membrane potential, triggering both mitophagy and cell death in senescent cells (Hao et al, 2022; Peña-Blanco et al, 2020). This reflects the context-dependent regulation of mitochondrial biogenesis, where nutrient-rich and nutrient-deprived conditions trigger distinct signalling pathways with varying effects.

## Mitochondrial alterations in aged naïve and memory T cells

Although a comprehensive characterisation and understanding of T cell ageing remains to be explored, it is known that T cell subsets are impacted by their metabolism during ageing (Renkema et al, 2014; Wertheimer et al, 2014; Ron-Harel et al, 2018). For instance, aged naïve CD4+ T cells are not only reduced in number but also display dysfunctional one-carbon metabolism, the most induced metabolic pathway during the early stage of T cell activation. Interrupted one-carbon metabolism in naïve T cells leads to impaired proliferation and activation (Ron-Harel et al, 2016, 2018). In contrast, the activation of naïve CD8+ T cells has been shown to rely on mitochondrial biogenesis during early activation to generate effector cells and cytokines, demonstrated using both pharmacological and genetic tools that inhibit mitochondrial translation (Fischer et al, 2018; Lisci et al, 2021). Although evidence of the changes in mitochondrial translation in naïve T cells during ageing is lacking, increased mitochondrial oxidative stress and decreased mitochondrial membrane potential imply their dysfunctional status (Fischer et al, 2018). In fact, suboptimal proliferation and the development of apoptosis-prone phenotype in naïve CD8+T cells in the context of ageing have already been reported by others. These changes are associated with excessive fatty acid uptake and lipid storage, resulting in dysfunctional fatty acid oxidation during the activation of naïve T cells. The molecular drivers and correlation to mitochondrial dysfunction remain unclear (Nicoli et al, 2022).

Like naïve T cells, metabolic reprogramming is also affected in memory T cells during ageing. In humans over 65 years of age, higher OXPHOS, ROS production, and ATP production are observed in CD4+ T memory cells compared to young (<35 years of age) individuals. This is accompanied by upregulated carnitine palmitoyltransferase 1A (CPT1a), which catalyses the transfer of the long-chain acyl group in acyl-CoA ester to carnitine, thus allowing fatty acids to enter the mitochondrial matrix for oxidation, resulting in higher fatty acid oxidation. This may affect the lipid storage needed for their long-term survival (Yanes et al, 2019). Although fatty acid metabolism has also been shown to support CD8+ T memory development (O'Sullivan et al, 2014), it remains unclear whether CD8+T memory cells share similar mitochondrial alterations. Metabolomic profiling of plasma from young and older healthy donors revealed distinct baseline metabolic signatures in response to influenza vaccination. Notably, young high responders (HRs) exhibited increased levels of upstream tryptophan metabolites such as L-tryptophan, 5-hydroxy-L-tryptophan (5-HTP), and indoleacetic acid at 28 days post-vaccination, a pattern absent in older subjects. This observation aligns with established evidence that shunting of tryptophan metabolism suppresses T cell responses

(Seo and Kwon, 2023). The preservation of immunomodulatory upstream tryptophan metabolites in young HRs suggests maintained tryptophan bioavailability, potentially supporting T cell responses driven by antigen presentation and interferon signalling. In contrast, older adults exhibit elevated triacylglycerols (TAGs) consumption, which are related to T memory and regulatory T cell (Treg) responses, suggesting older adults may rely on more T memory and Treg responses during influenza vaccination (Chou et al, 2022).

## Mitochondrial alterations in age-related T cell subsets

The ageing immune system undergoes profound changes beyond the decline of naïve and memory T cell function. As thymic output wanes and the naïve T cell pool diminishes, the body must compensate for both this loss and the constant immune challenges posed by lifelong antigen exposure. Simultaneously, inflammageing creates a hostile microenvironment, prompting the immune system to adapt by generating specialised T cell subsets like T virtual memory cells (TVM), terminally differentiated effector-memory cells that re-express CD45RA (TEMRA), and T ageing-associated (Taa) cells (Møller et al, 2022). TVM cells emerge due to homoeostatic proliferation and exhibit features of memory T cells, yet are antigen-naïve (Hussain and Quinn, 2019). Unlike conventional naïve T cells, aged CD8+ TVM cells exhibit marked functional decline, adopting a SASP phenotype and accumulating DNA damage markers like γH2AX (Quinn et al, 2018). Intriguingly, these cells preserve metabolic flexibility, maintaining glycolytic activity and spare respiratory capacity (SRC) — a critical reserve of mitochondrial ATP production during energy surges — at levels comparable to young naïve T cells (Borsa et al, 2021; Quinn et al, 2020). This suggests that TVM cells may represent an adaptive concession to sustain immune function in ageing. However, this adaptation comes with compromises. Aged TVM cells display excessive mitochondrial fusion, impairing organelle turnover and reducing metabolic plasticity. Paradoxically, they also upregulate ETC activity and SRC, likely driven by elevated Pgc-1α and suppressed mitophagy (↓Atg101, ↓Ulk1) (Quinn et al, 2018, 2020). This imbalance, where mitochondrial biogenesis outpaces degradation, may explain their dual phenotype: energetically competent but functionally senescent.

Another senescent/exhausted subset, TEMRA, arises under repeated antigen exposure and chronic infections in humans (Jain et al, 2023). These cells are characterised by reduced proliferative capacity and diminished mitochondrial function while maintaining their cytotoxicity. Higher ROS production and reduced mitochondrial membrane potential are observed in TEMRAs compared to other subsets of CD8+ T cells (Henson et al, 2014). CD4+ TEMRAs, on the other hand, accumulate less with age, perhaps due to their higher mitochondrial mass; the mechanisms remain unclear (Strickland et al, 2023; Callender et al, 2020). Recent discovery of the Taa population further completes the T cell ageing picture. Taa cells become prevalent in older people and are characterised by the expression of granzyme K (GZMK), exhaustion marker-programmed cell death protein 1 (PD-1) and the transcription factor TOX (Mogilenko et al, 2021). While the origin of Taa is not fully understood, it has been shown that Taa can influence systemic ageing by inducing a SASP phenotype in aged fibroblasts, correlating with raised inflammatory cytokines

interleukin 6 (IL-6), interleukin 8 (IL-8), and tumour necrosis factor-alpha (TNF-α) in the plasma. Furthermore, a similar phenotype is observed when Tfam is genetically deleted in murine T cells, where impaired mitochondrial function in this cellular compartment results in systemic organismal ageing (Desdín-Micó et al, 2020). These findings reveal a feedback loop in ageing immunity: as T cells lose their functional integrity, their metabolic dysregulation and inflammatory signals exacerbate tissue deterioration, which in turn puts further strain on the immune system. The key problem seems to be dysfunctional mitochondria, which different T cell subsets accumulate to a variable degree, yet they are universally disruptive. Deciphering how mitochondrial pathways govern T cell behaviours could unlock targeted strategies to break this loop, potentially restoring immune resilience and mitigating age-related decline.

## Drugs and interventions that target mitochondrial products

As mitochondrial health is closely linked to ROS production, multiple antioxidant strategies have been investigated. The antioxidant drug N-acetylcysteine (NAC) and vitamin C were given to aged mice during vaccination and resulted in improved long-lived immune memory formation (Meryk et al, 2020). This builds on existing evidence that in vitro use of NAC can reduce ROS production and reverse telomere length reduction in primary T cells from aged donors (Sanderson and Simon, 2017). These findings have now been adapted in a phase I clinical trial together with chimeric antigen receptor T cell (CAR-T) therapy for the treatment of non-Hodgkin's lymphoma (NCT05081479). In response to the rapid ROS rise following T cell activation, an endogenous antioxidant response is triggered, marked by glutathione (GSH) production, a major detoxification agent that scavenges ROS (Mak et al, 2017). Endogenous GSH levels are reduced during terminal differentiation of T cells, which conversely could be prevented by antioxidants that preserve a T stem cell memory phenotype with better adoptive transfer and killing potential (Pilipow et al, 2018). Other quinone-based antioxidants, for example, co-enzyme A, and co-enzyme Qs, were also shown to be important in maintaining T cell anti-tumour functions (St. Paul et al, 2021; Reina-Campos et al, 2023; Hidalgo-Gutiérrez et al, 2021).

Mitochondrial biogenesis, aimed at improving mitochondrial health, has been explored in preclinical and clinical settings (Kurmi et al, 2023). One promising strategy involves boosting NAD+-dependent sirtuins, particularly SIRT1, which declines in aged T cells and other tissues (Jeng et al, 2018; Xu et al, 2020). NAD+ precursors like nicotinamide riboside (NR) reverse inflammatory and ageing phenotypes in T cell-specific TFAM-knockout mice (Desdín-Micó et al, 2020). Furthermore, genetic disruption of de novo NAD+ synthesis (e.g. via the kynurenine pathway) impairs anti-tumour T cell responses (Wan et al, 2023), although there is limited evidence on whether this increases cancer susceptibility during ageing. Clinically, NR and NMN (Nicotinamide mononucleotide) supplementation are being tested for metabolic and ageing-related disorders (Akasaka et al, 2023; Dollerup et al, 2018). It is also noteworthy that metabolic dysfunction in CD8+ T cells, driven by the immunosuppressive microenvironment of chronic lymphocytic leukaemia (CLL), contributes to CAR-T therapy resistance (Van Bruggen et al, 2019). By co-expressing an inhibition-resistant PGC-1α variant with anti-EGFR

**Mitochondrial ROS production and biogenesis**

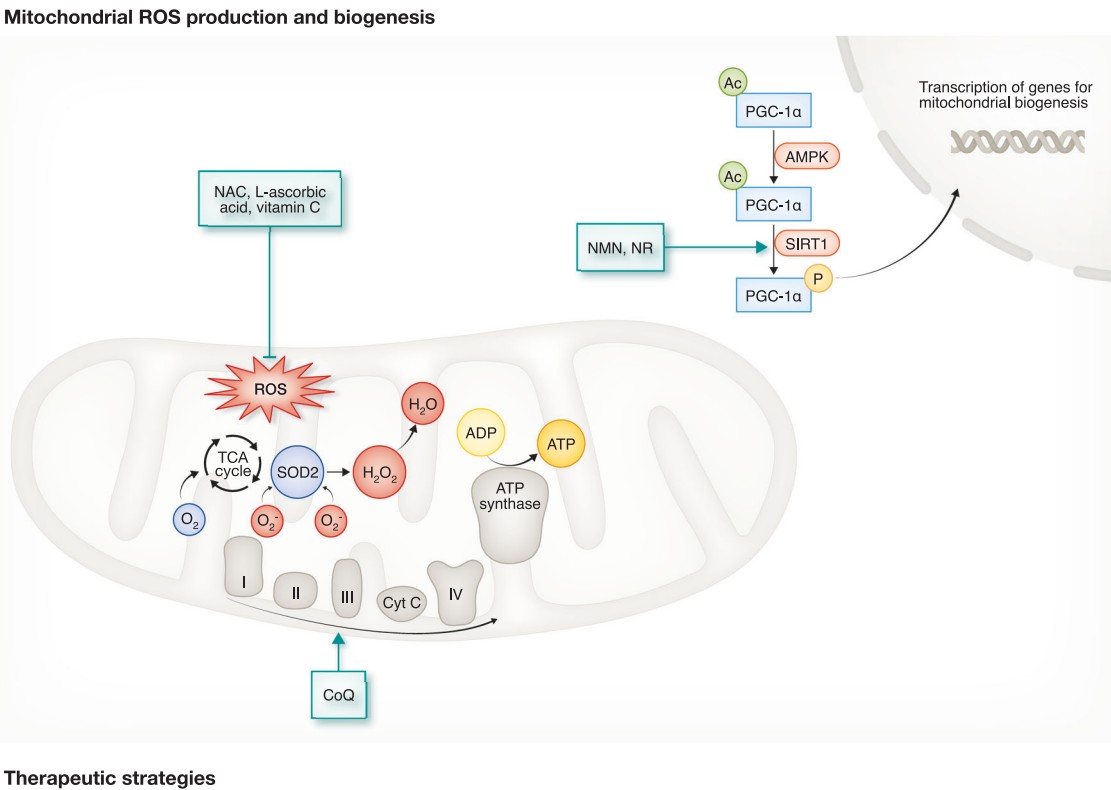

**Figure 2. ROS production and mitochondrial biogenesis and its therapeutic potentials.**

Reactive oxygen species (ROS) are generated and scavenged during mitochondrial metabolism as part of ATP production. At basal levels, ROS play a physiological role in promoting mitochondrial biogenesis and signalling. However, with ageing, ROS levels can exceed a critical threshold, leading to oxidative damage and impaired mitochondrial function. This contributes to less efficient mitochondrial biogenesis and dysfunctional energy metabolism in T cells. Both increased ROS and impaired biogenesis represent potential therapeutic targets, which may be addressed using antioxidants and NAD+ precursors, respectively. Green arrows or inhibition marks with solid line represent drug effects supported by experimental evidence.

CARs, T cell metabolic fitness and tumour control are enhanced (Lontos et al, 2023), revealing a promising strategy to synergise metabolic reprogramming with adoptive cell therapy (Fig. 2).

Mitochondrial translation influences T cell function in a number of ways. For example, experimentally induced fever enhances mitochondrial translation in CD8+ T cells, boosting their anti-tumour responses, which is inhibited by the translation inhibitor tigecycline (O'Sullivan et al, 2021). Similarly, genetic deletion of the deubiquitinase USP30 or pharmacological inhibition of mitochondrial translation (e.g. with doxycycline or chloramphenicol) impairs cytotoxic granule formation and T cell killing (Lisci et al, 2021). However, how mitochondrial translation behaves during ageing in T cells requires more investigation.

In summary, managing mitochondrial byproducts such as ROS, and enhancing mitochondrial biogenesis hold promise with the goal to improve aged T cell functions and promote systemic health. In the following sections, we will focus on how different cell biological pathways can promote mitochondrial homoeostasis, are regulated by T cell ageing and whether their therapeutic modulation can counteract immunosenescence.

## Mitophagy to delay ageing

One way to remove unwanted mitochondria is by delivering the mitochondria to the lysosomes via autophagy, a process called mitophagy (Ganley and Simonsen, 2022). Before mitochondria can be taken up by autophagosomes, they are thought to require fragmentation. Piecemeal mitophagy is the removal of damaged macromolecules, where dysfunctional mitochondrial macromolecules are shuffled into a specific area, enabling the autophagosome to pinch off this portion (Ganley and Simonsen, 2022). Accumulation of dysfunctional mitochondria is thought to be a consequence of an age-related decline in mitophagy, but to date, little is known about this process in ageing T cells (Chen et al, 2020). Outside the context of ageing, depolarised mitochondria have been shown to accumulate as a result of decreased mitophagy activity in tumour-infiltrating T lymphocytes. Consequently, these cells display the functional, transcriptomic and epigenetic characteristics of terminally exhausted T cells (Yu et al, 2020).

Mitophagy shares its molecular machinery with autophagy (Ganley and Simonsen, 2022). However, what differentiates

mitophagy from autophagy at the molecular level are the receptors that enable the autophagosomal machinery to recognise damaged or unwanted mitochondria. Over the last two decades, several receptors have been described in mammalian cells. Here, we will focus on the three main mitophagy pathways. The best described is the Pink-Parkin pathway, in which Pink1 senses mitochondrial transmembrane potential loss and recruits Parkin, an E3 ubiquitin ligase, to the damaged mitochondria (Narendra and Youle, 2024). This pathway is engaged upon starvation or when mitochondria are damaged. The PINK1/Parkin pathway is found not to be functional in human peripheral mononuclear cells, of which T lymphocytes are one subset. The mitophagy receptors used in the PINK1/Parkin pathway are NDP52, OPTN and TAX1BP1 (Narendra and Youle, 2024). Hypoxia-induced mitophagy operates through the FUN14 domain-containing protein 1 (FUNDc1) pathway. FUNDc1 also serves as a mitophagy receptor when mitochondria are uncoupled, or when paternal mitochondria are removed in C. elegans (Chen et al, 2020). FUNDc1 is an outer mitochondrial membrane protein with an LC3-interacting domain (LIR). One of the earliest mitochondrial receptors that were discovered are BNIP3 and NIX. NIX was shown to remove mitochondria from red blood cells during their differentiation (Schweers et al, 2007; Sandoval et al, 2008) and has possible roles in cancer, in heart myocytes and in macrophages (Chen et al, 2020).

## Mitophagy during ageing

There is some evidence that adequate and timely removal of mitochondria by mitophagy is key to a long healthspan and lifespan. This evidence is primarily based on C. elegans, where knocking out mitophagy receptors, recapitulates the effect of ageing on mitochondrial mass or shortens their lifespan when all three worm homologues of Pink, Parkin and Nix are knocked out (Palikaras et al, 2015). However, studies using mammalian cells during ageing show contradictory results (Bakula and Scheibye-Knudsen, 2020). Indeed, a recent study reported no obvious decline in mitophagy in mice across all brain tissues with age, but rather region-specific dynamics (Rappe et al, 2024). Another study reports that while autophagy levels decline, mitophagy levels remain stable in several tissues analysed (Jimenez-Loygorri et al, 2024). This field requires more evidence from other tissues and better quantitative in vivo tools. For instance, the detection of mitochondrial proteins in the autophagic cargo of aged cells would provide direct evidence of mitophagy's role in healthy ageing, which is still lacking despite the development of novel tools to identify autophagosomal cargo (Zhou et al, 2022; Zellner et al, 2021).

## Mitophagy impacts T cells and T cell ageing

Numerous studies show that T cells must undergo mitophagy during their differentiation, upon activation or to ensure survival. The earliest studies using knock-out mice of key autophagy genes targeted to T cells showed an excess of mitochondria in T cells (Pua et al, 2009; Mortensen et al, 2010). Similarly, memory T cells are not maintained in a model of Atg7 deletion specifically in T cells and accumulate damaged mitochondria (Puleston et al, 2014). A complementary study found a similar defect in memory CD8+ T cells when mice were knocked out for Atg5 in effector T cells,

accompanied by a change in lipid metabolism (Xu et al, 2014). Loss of Atg5 has also been reported to promote increased CD8+ T cell anti-tumour activity, which the authors suggest might be caused by their biased differentiation towards an effector-memory phenotype. Although evidence of enhanced effector functions in Atg5-deficient cells is compelling, which is accompanied by increased glucose metabolism and epigenetic regulation of effector and metabolic target genes, the role of autophagy and mitophagy was not directly addressed, neither was the distinction of short-lived effector cells and effector-memory T cells. Indeed, this work provides further evidence on the key role of autophagy in the long-term maintenance of stem-like quiescent T cells (Devorkin et al, 2019). Deletion of Atg5 in T regulatory (Treg) cells shows enhanced DNA damage, excessive mtDNA, and reduced survival of Tregs (Alissafi et al, 2020). Many other studies link dysfunctional mitochondria and reduced mitophagy with cell death. For example, activation-induced cell death (AICD) can only go ahead when general macroautophagy is inhibited (Corrado et al, 2016). Autophagy-defective human naive CD4+ T cells were susceptible to apoptosis, and here, while mitophagy was not quantified, ULK1-deficient CD4+ T cells were refractory to cell death when intracellular ROS could be reduced (Watanabe et al, 2014). In 2010, Macian and colleagues used electron microscopy to convincingly demonstrate that mitochondria are excluded from autophagosomes in effector T cells (Hubbard et al, 2010). Aside from this study, a general drawback of early research was the initial challenge of studying mitophagy isolated from general autophagy. One could speculate that the deletion of key autophagy genes could indirectly lead to changes in mitochondria, for example, via increased mitochondrial biogenesis or preferential survival of cells that retain mitochondria and may not directly stem from deleterious degradation.

Once the first mitophagy receptors were identified, its significance in T cells could be clarified in a more definitive manner. For example, when the mitophagy receptor NIX was specifically knocked out in T cells, a defect in the formation and maintenance of memory T cells was found, recapitulating the earlier studies with Atg5 or Atg7 and confirming that NIX is a key mitophagy receptor in T cells (Gupta et al, 2019). It would be interesting to monitor these mice for a longer period to observe how their phenotype evolves with age. Other mitophagy receptors revealed controversial results. Pink and Parkin are upregulated in memory T cells: Parkin suppressed VDAC-1-dependent apoptosis and NIX counteracts ferroptosis resulting from impaired mitophagy (Franco et al, 2023). While in one study the PINK1/Parkin system was found not to be functional in human peripheral blood mononuclear cells (Bradshaw et al, 2021), another study found that the knockout of Pink rather promotes the differentiation into Th1 cells in mice (Mei et al, 2023). Another study compared mitophagy levels using a mitophagy dye and found that naïve CD4+ T cells had the highest level of mitophagy followed by memory CD4+ cells and Tregs; however CD8+ T cells altogether showed very low levels of mitophagy. This reveals that these T cell subsets are very different (Liang et al, 2022). A comparison between young and old mice would have been a great addition to this study.

Very few studies show that a decline in mitophagy contributes to the ageing of T cells. One study measured mitophagy in human CD4+ T cells. They found a significantly higher number of autophagosomes with many containing undegraded mitochondria

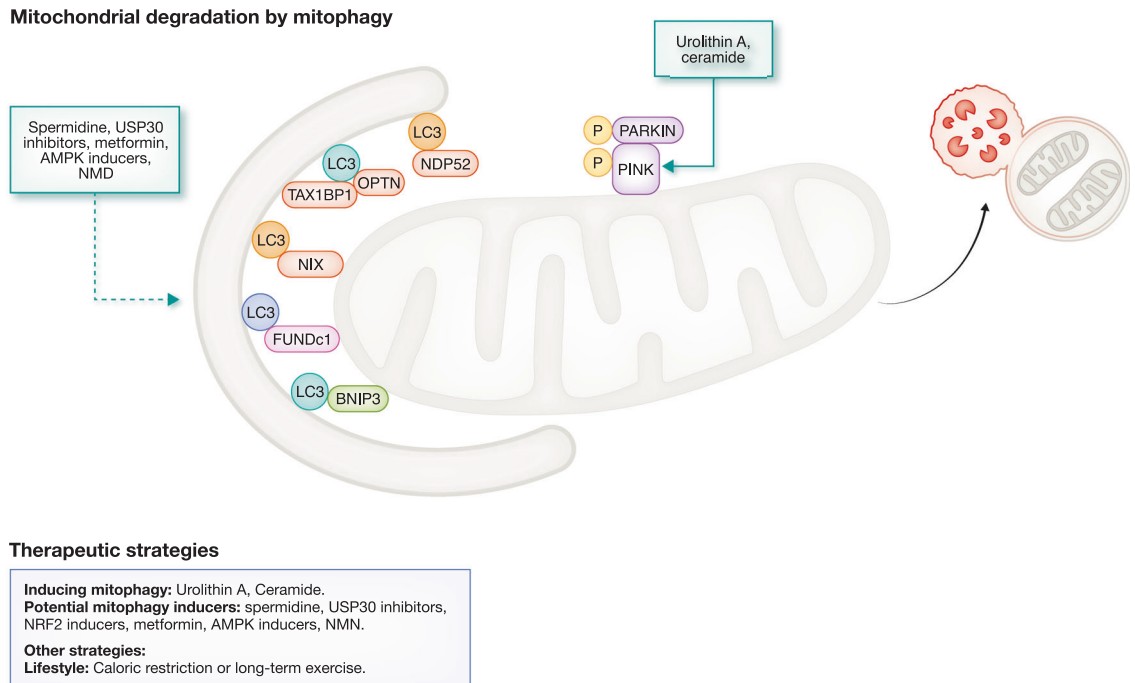

**Therapeutic strategies**

**Inducing mitophagy:** Urolithin A, Ceramide.
**Potential mitophagy inducers:** spermidine, USP30 inhibitors, NRF2 inducers, metformin, AMPK inducers, NMN.
**Other strategies:**
**Lifestyle:** Caloric restriction or long-term exercise.

**Figure 3. Mitochondrial degradation and its therapeutic potentials.**

Mitochondrial clearance occurs primarily through macroautophagy, mediated by key mitophagy signals: the PINK1/Parkin pathway, mitophagy receptors: FUNDc1, NIX and BNIP3. Enhancing autophagy via pharmacological interventions or lifestyle modifications represents a promising therapeutic approach to restore mitochondrial health. Green arrow with solid line represents drug effects supported by experimental evidence, whereas green arrow with dashed lines indicates proposed therapeutic potential.

in older adults as compared to younger, suggesting defective mitochondrial turnover by autophagy (Bektas et al, 2019). Whether this causes an aged phenotype is difficult to discern, without reversing the ageing phenotype with an autophagy/mitophagy-inducing compound.

## Drugs and interventions that target mitophagy

Several studies focusing on mitophagy as a therapeutic target already exist. While many have been conducted in young organisms, some have also included the aged. One study performed in young mice used the mitophagy-inducing compound Urolithin A to improve T memory stem cell formation by inducing Pink1-dependent mitophagy (Denk et al, 2022). We treated old mice with spermidine and found that it can recover T cell memory responses, an effect that is not observed in the absence of autophagy (Puleston et al, 2014). Together with the studies showing a surplus of mitochondria in Atg7 knockout CD8+ T cells (Puleston et al, 2014), this suggests that spermidine may have an effect on mitophagy. Alternatively, spermidine may turn on mitochondrial translation, as shown in macrophages (Puleston et al, 2019). Similarly, another study found that NMN might act via mitophagy to prevent senescence by promoting mitochondrial homoeostasis in tumour-infiltrating CD8+ T cells (Ye et al, 2024). In contrast, inhibiting ceramide-dependent mitophagy can improve the age-related dysfunction of anti-tumour T cells; however, they did not measure levels of mitophagy/autophagy in the presence of flux inhibitors (Vaena et al, 2021).

Lifestyle interventions such as long-term exercise and calorie restriction have shown promise in alleviating the phenotype of ageing in other tissue types, but clinical trials demonstrating an effect on mitophagy in humans are still lacking. However, clinical trials with a USP30 inhibitor, or modulators of the NRF2/KEAP pathway — a pathway that regulates cellular response to oxidative stress, metformin or other AMPK activating compounds are underway, as is the screening for more specific mitophagy-inducing drugs (Picca et al, 2023). This is a promising area which will bring us closer to the rejuvenation of T cell responses (Fig. 3).

## Asymmetric inheritance of mitochondria

Another way for long-lived T cells to remove mitochondria is by passing them on to the shorter-lived daughter cells during division (segregation), which is achieved by asymmetric cell division (ACD). Although this is a conserved mechanism observed across different species and cell types, its physiological relevance is still a field of much debate. And certainly, when it comes to mitochondria, the consequences of asymmetric distribution in terms of health and cell function remain unclear.

The asymmetric inheritance of cell cargoes during mitosis is amongst the mechanisms that contribute to early fate decision during CD8+ T cell differentiation (Chang et al, 2007). In T cells, ACD is particularly important for the formation and maintenance of long-lived memory (Chang et al, 2007, 2011; Borsa et al, 2019). ACD relies on high-affinity TCR stimulation in the context of an

immune synapse. Weak TCR-binding due to low-affinity antigen presentation or the absence of integrins that stabilise the interaction between the T cell and the antigen-presenting cell (APC) results in poor antigen-specific memory in vivo (Gräbnitz et al, 2023; King et al, 2012). A mother cell undergoing ACD gives rise to two daughter cells that inherit distinct cellular content, which is represented by several layers of asymmetry, including the differential expression of surface markers, transcription factors and divergent metabolic activity and translation profiles (Chang et al, 2011; Pollizzi et al, 2016; Verbist et al, 2016). First-daughter cells following CD8+ T cell mitoses already exhibit divergent transcriptomes, which further contributes to their fate bias (Borsa et al, 2019; Quezada et al, 2023; Metz et al, 2015; Liedmann et al, 2022). Similar to other cell types that rely on ACD to maintain stemness, such as haematopoietic stem cells (Loeffler et al, 2019), whose ability to divide asymmetrically is compromised with ageing (Florian et al, 2018), naïve CD8+ T cells from older mice also lose this ability. This jeopardises their maintenance as quiescent and long-lived cells (Borsa et al, 2021). Cells that retain their ability to undergo ACD during ageing, such as TVM, also maintain their potential to generate diverse progenies containing both memory cells and effector cells. Thus, ACD seems to be sufficient to preserve stemness even in scenarios where cell-extrinsic features, such as an ageing environment, would not favour that fate.

## Mitochondria as drivers of asymmetric fates

The asymmetric layers observed during ACD, root from early synaptic events, as the immune synapse between the T cell and the APC functions as an anchor point for the establishment of a polarisation axis that is maintained throughout cell division. Interestingly, mitochondria are amongst the cell cargoes polarised towards the synapse. There, mitochondria are important in regulating calcium signalling and impact on immune synapse architecture (Quintana et al, 2007; Baixauli et al, 2011; Quintana et al, 2011). Cytotoxic CD8+ T cells (CTLs) lacking mitochondria exhibit dysfunctional immune synapses and impaired T cell killing (Lisci et al, 2021). However, it is not yet clear whether early polarisation events occurring at the immune synapse have a direct impact on the (post-)mitotic inheritance of heterogeneous mitochondrial pools by T cell progenies.

Mitochondria have their own DNA and ribosomes, and the total pool of mitochondria in a cell mostly results from a balance of mitochondrial biogenesis and mitophagy. Upon proliferation, cells can also partition their mitochondrial content between daughter cells, which allows mitochondria segregation and the inheritance of heterogeneous organelle pools. From the perspective of ACD (segregation), different reports investigating whether mitochondria are asymmetrically inherited during mitosis generated conflicting results (Pollizzi et al, 2016; Verbist et al, 2016; Adams et al, 2016; Emurla et al, 2021). Human mammary epithelial cells (hMECs) can asymmetrically apportion old mitochondria during cell division. The progeny inheriting old mitochondria undergoes terminal differentiation and loses the potential to self-renew, while the progeny that does not inherit old organelles remains stem-like (Katajisto et al, 2015). Fate divergence results from distinct metabolic programmes driven by old and young mitochondrial compartments. Old mitochondria are sites of oxidative phosphorylation, which drives hMECs differentiation, while young

mitochondria have high PPP, which supports cell stemness by promoting de novo purine biosynthesis and redox balance (Döhla et al, 2022). In CD8+ T cells, several reports demonstrated that daughter cells are endowed with distinct metabolic profiles following ACD: the daughter cell that is proximal to TCR signalling exhibits higher uptake of nutrients, mTOR activity and cMyc expression, which fuels glycolysis and differentiation into effector cells (Pollizzi et al, 2016; Verbist et al, 2016; Liedmann et al, 2022). As mitochondria are central to T cell metabolism, it remained to be investigated whether these functional differences stemmed from mitochondrial diversity.

## Asymmetric inheritance of mitochondria in T cells

We have recently addressed the aforementioned open question and reported that unequal inheritance of different pools of mitochondria in CD8+ T cells results in asymmetric fates. We took advantage of a mouse model that shares the technology of sequential labelling of mitochondrial structures and their discrimination by chronological age, used to study hMECs, but now allowing the investigation of primary cells directly isolated from living tissue. We demonstrate that cells capable of high mitochondrial turnover, thus not accumulating old/damaged organelles, can maintain quiescence. Furthermore, when used in adoptive cell transfer experiments, these cells exhibit better survival and re-expansion potential. In contrast, cells that maintain old mitochondria for longer periods are more likely to become effector cells and strongly rely on glycolysis and 1C metabolism to meet their metabolic demands. Interestingly, autophagy is required for the emergence of cells devoid of old mitochondria by impacting not just degradation but also the unequal segregation of these organelles. In autophagy-deficient cells, aged mitochondria are inherited symmetrically. Thus, one can speculate that in T cells from aged individuals, when autophagy is impaired, asymmetric segregation of mitochondria would also be negatively impacted, which would contribute to less diverse progenies and diminished memory potential. Nevertheless, this evidence suggests that fate commitment is impacted by early events of mitochondrial inheritance that shape T cell metabolism (Borsa et al, 2024). It remains unclear whether the distinct metabolic profiles observed may drive epigenetic changes that might sustain divergent fate trajectories. Furthermore, as mitochondria superoxide is expressed in older mitochondria domains and ROS has been associated with telomere shortening, it remains to be addressed whether the long-term replicative capacity of daughter cells is determined by early events of mitochondrial inheritance.

## Drugs and interventions that target asymmetric cell division

ACD in both young and old T cells can be boosted by short-term mTOR inhibition (Borsa et al, 2021, 2019). Low-dose and transient rapamycin treatment after T cell activation but prior to T cell division can increase ACD rates and restore this ability in progenitor exhausted CD8+ T cells (TCF1+) and naïve CD8+ T cells from aged individuals. This leads to T cell rejuvenation and improved memory potential. Mechanistically, rapamycin has been shown to strengthen diffusion barriers that are formed in the endoplasmic reticulum (ER) of mitotic CD8+ T cells and that are required to maintain asymmetries during cell division. It remains to be investigated whether the segregation of cellular components not

attached to ER membranes is also modulated by transient mTOR inhibition (Emurla et al, 2021).

Autophagy in T cells can also be induced pharmacologically by spermidine and is linked to better T cell immune responses in the elderly. As autophagy is required to promote ACD in CD8+ T cells, it is possible that autophagy modulation could impact the asymmetric inheritance of different mitochondrial pools. It is also plausible to speculate that the selective partitioning of mitochondrial populations relies on their architecture and/or the expression of adaptors required by their cytoskeleton-dependent transport. A recent report showed that ACD can also be important in the context of CAR-T cells to promote the generation of diverse progenies (Lee et al, 2024). As ACD modulation can be successfully done in CD8+ T cells isolated from human PBMCs (Borsa et al, 2019), modulation of mitochondrial inheritance emerges as a potential tool to improve the outcomes of adoptive cell transfer therapies by promoting the maintenance of long-lived cells while not compromising their needed effector functions (Fig. 4).

## Mitochondrial transfer

The role of mitochondria in sustaining cellular and tissue health is crucial, as they function as central signalling hubs that link

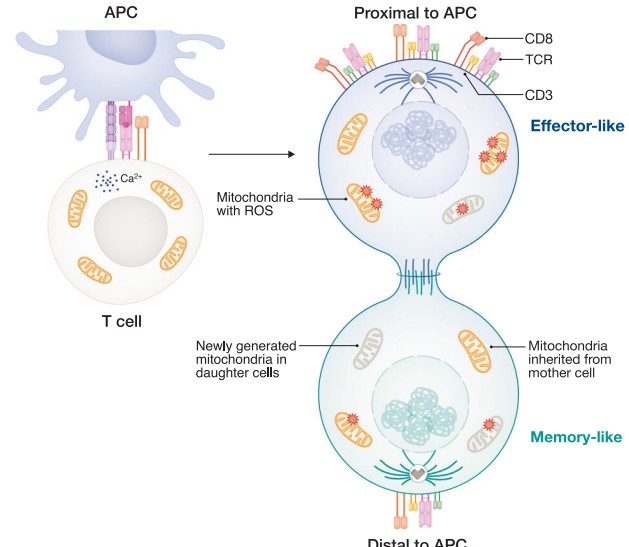

**Asymmetric inheritance of mitochondria during mitosis**

**Therapeutic strategies and perspective**

**Inducing ACD:** Rapamycin
**Perspectives:** Implying outcomes of CAR-T cell therapy and vaccine responses

**Figure 4. Asymmetric inheritance of mitochondria and its therapeutic potentials.**

Following activation by antigen-presenting cells (APCs), T cells form polarity and undergo mitosis with a proximal and distal side depending on the contact with APCs. During this process, mitochondria can be inherited through asymmetric cell division (ACD). Selective mitochondrial inheritance through ACD offers a potential mechanism for clearing damaged mitochondria. Transient rapamycin treatment enhances ACD efficiency. Improving ACD also holds promise for improving CAR-T cell therapy and vaccination response.

molecular systems within individual cells all the way to communication networks across diverse cell types. This connectivity underscores the inherently social nature of mitochondria. By being transferred from one cell to another or parts thereof, mitochondria mediate extracellular and intercellular communication. The concept of horizontal mitochondrial transfer — where whole functional mitochondria are exchanged between cells — was long seen as theoretical until recent breakthroughs provided in vivo evidence (Brestoff et al, 2025). For example, transmissible cancer models in dogs showed host-cancer cell mitochondrial transfer, providing the first experimental proof of intercellular mitochondrial exchange (Rebbeck et al, 2011). Mitochondrial transfer has since been observed in a variety of settings, including mouse models of neuroinflammation (Peruzzotti-Jametti et al, 2021; English et al, 2020), cancer (An et al, 2015), lung diseases (Li et al, 2014), as well as in systems such as adipocytes transferring mitochondria to heart cells (Crewe et al, 2021), in lymphocytes (Court et al, 2020; Harada et al, 2022), and platelets exchanging mitochondria with cells in the blood vessel wall (Levoux et al, 2021). Tracking mitochondrial transfer in humans has proven challenging, with limited studies available. Bulk and single-cell RNA sequencing in a study of cancer patients identified mitochondrial mutations and gene expression signatures consistent with the exchange of mitochondria from T cells to cancer cells. Predicted mitochondrial recipient cells exhibited specific markers associated with energy generation and markers related to cytoskeleton regulation (Zhang et al, 2023).

### Mechanisms of mitochondrial transfer

Mitochondrial transfer primarily occurs through two mechanisms: (i) the release of vesicles containing mitochondrial components like mtDNA, proteins, or metabolites and (ii) direct cell-to-cell mitochondrial transfer. For the latter, mitochondria can be exchanged between cells through various mechanisms such as cell fusion (Qiao et al, 2024), GAP junctions (Li et al, 2019; Islam et al, 2012; Fahey et al, 2022) and tunnelling nanotubes (TNTs). Mitochondrial transfer has been found between neural stem cells and ischaemic neurons (Capobianco et al, 2024), fibroblasts or T cells to cancer cells (Saha et al, 2022; Qiao et al, 2024), and mesenchymal stem cells to epithelial cells (Yao et al, 2018). For the first, transfer occurs through non-contact mechanisms such as cell-free mitochondria (Stephens et al, 2020; Nicolás-Ávila et al, 2020; Boudreau et al, 2014) and extracellular vesicles, including exosomes and mitochondrial-derived vesicles (MDVs), which are taken up through caveolae-mediated endocytosis (Torralba et al, 2018; Hayakawa et al, 2016; Crewe et al, 2021). In immune cells, mechanisms like trogocytosis may also actively facilitate mitochondrial exchange (Joly and Hudrisier, 2003). Mitochondrial dynamics within donor cells and their energy production capacities are believed to influence their transfer. Cytoskeletal support also plays a critical role in mitochondrial trafficking. When mitochondria are damaged, recipient cells often degrade them through transmitophagy as observed in retinal ganglion cells and cardiac tissue (Davis et al, 2014; Melentijevic et al, 2017). By contrast, functional mitochondria appear essential for effective exchange and integration (Levoux et al, 2021), suggesting that exogenous mitochondria can either fuse with or be degraded by recipient cells, depending on their status.

## Mitochondrial transfer from and to T cells

Unidirectional transfer has been observed from T cells to cancer cells, as well as from bone marrow stromal cells (BMSCs) and CD8+ T cells, facilitated by TNTs (Saha et al, 2022; Baldwin et al, 2024). In this process, a single nanotube extending from a cancer cell can connect to multiple immune cells in a chain, forming several contact sites along the lymphocyte's cell membrane. This mechanism is similar to the intercellular mitochondrial transfer previously reported from mesenchymal stem cells (MSCs) to epithelial cells (Ahmad et al, 2014). During mitochondrial exchange from MSCs to CD4+ T cells, neither TNTs, gap junctions, macropinocytosis, nor hemi-channels were found to be important for the process. Only the inhibition of extracellular vesicles (EVs) effectively blocked mitochondrial transfer (Court et al, 2020). However, limitations exist; these transferred mitochondria from murine cells into human T cells trigger a signalling cascade but are likely degraded due to incompatibilities in species-specific mitochondrial DNA replication machinery (Court et al, 2020). In summary, the precise molecular mechanisms controlling mitochondrial release and uptake remain elusive, likely differing across cell types and specific cellular contexts. It remains unclear whether mitochondrial transfer plays a role in alleviating cellular ageing physiologically, and its potential for therapeutic use remains to be addressed fully.

## Mitochondrial transfer in T cell function and therapeutic potential

In vivo mitochondrial delivery has shown promise in restoring cellular functions in various conditions, including hepatic, pulmonary, renal, and cardiovascular pathologies (Shi et al, 2017; Nakai et al, 2024; Shi et al, 2018; Sun et al, 2015; Cowan et al, 2016), with tools like FluidFM (Gäbelein et al, 2022) enabling precise manipulation of mitochondria for their extraction and injection into living cells. However, challenges remain, particularly regarding potential immune responses triggered by allogeneic mitochondrial transplantation, which may activate mitophagy due to species incompatibilities and the immune system due to antigen presentation of foreign mitochondrial peptides (Ramirez-Barbieri et al, 2019; Pollara et al, 2018; Lin et al, 2019). Beyond functional recovery, mitochondrial transfer may influence cell reprogramming. For instance, platelets can drive metabolic remodelling in mesenchymal stem cells through mitochondrial transfer, enhancing their regenerative capacity (Levoux et al, 2021). Although limitations remain, especially in understanding the role of transferred mitochondria in differentiated cells, these advancements hold immense potential for sustainable cell therapies.

T cell-based therapies that strengthen immune responses through metabolic reprogramming are rapidly advancing in the field. Various techniques target mitochondrial pathways to boost T cell expansion, cytotoxic activity, and longevity in therapeutic contexts (Guo et al, 2021; Yang et al, 2016). Among these, CAR-T cell therapies stand out; while metabolic interventions have enhanced CAR-T cell function in the short term, maintaining these benefits over the long term remains challenging (Fultang et al, 2020; Pilipow et al, 2018). Mitochondrial transfer — whether through whole organelles or mitochondrial components — holds transformative potential for addressing mitochondrial dysfunction and metabolic deficiencies within T cells. This process can notably enhance the metabolic flexibility of T cells, which is crucial for their activation, persistence, and function. It has previously been reported that mitochondrial transfer influences the functionality of T cell subsets. For instance, murine MSC-derived mitochondria, when transferred to human CD4+ T cells, have been shown to induce Treg differentiation (Court et al, 2020). A 2D tissue culture with a co-culture transwell system has recently emerged as an effective platform for intercellular mitochondrial transfer from bone marrow stromal cells (BMSCs) to CD8+ T cells (Baldwin et al, 2024). This mitochondrial exchange enhances the anti-tumour immunity of CD8+ T cells by promoting their survival, expansion, and cytotoxic functions within the tumour microenvironment. Mechanistically, CD8+ T cells receiving exogenous mitochondria display increased protein synthesis, reduced expression of exhaustion markers (PD-1, LAG3), and lower rates of apoptotic death (Baldwin et al, 2024). Antigen-activated aged T cells and anti-tumour T cells share notable dysfunctions: both display metabolic issues, decreased mitochondrial function, and increased expression of inhibitory receptors such as PD-1 and cytotoxic T-lymphocyte antigen 4 (CTLA-4), further diminishing immune activity (Zhang et al, 2020). In aged CD4+ T cells, the introduction of mitochondria from younger cells activates the expression of mitochondrial respiratory complexes, enhancing both mitochondrial respiration and glycolysis (though with a lower glycolytic reserve) to boost ATP production (Headley et al, 2024). This approach ultimately activates TCR signalling, leading to increased CD4+ T cell proliferation ex vivo and enhanced protection against infections in murine models (Headley et al, 2024).

It is essential to further explore the outcomes of the horizontal transfer of mitochondria. The transfer of mitochondria from mesenchymal stem cells (MSCs) to activated CD4+ T cells induces immunosuppression through the downregulation of the key Th1 transcription factor T-bet in the context of autoimmunity (Akhter et al, 2023). In contrast, the transfer of mitochondria from BMSCs to CD8+ T cells enhances the anti-tumour immune response (Baldwin et al, 2024). Additionally, cancer cells in the tumour microenvironment transfer mutated mitochondria to tumour-infiltrating lymphocytes (TILs), impairing their immune response by inhibiting mitophagy and inducing metabolic impairment and senescence (Ikeda et al, 2025). This discrepancy suggests either the cellular origin of mitochondria, or the activation/metabolic state of the recipient cell influences the reprogramming capabilities. Further development of in vitro and in vivo models is essential to evaluate mitochondrial transfer's long-term effects and ensure mitochondrial functionality post-transfer, facilitating exploration of the therapeutic benefits of this phenomenon in aged T cells (Fig. 5).

## Conclusions and perspectives

The maintenance of mitochondrial homoeostasis is considered central to cellular health and prevents ageing. Accordingly, the age-related changes occurring in T cells are closely associated with metabolic health, with a well-recognised direct link to mitochondrial health. Within the heterogeneity of the immune system, T cells have emerged as pivotal to organismal health. However, T cells are not the only cell type to have this effect. Other cell types that are well distributed throughout the body, can trigger similar

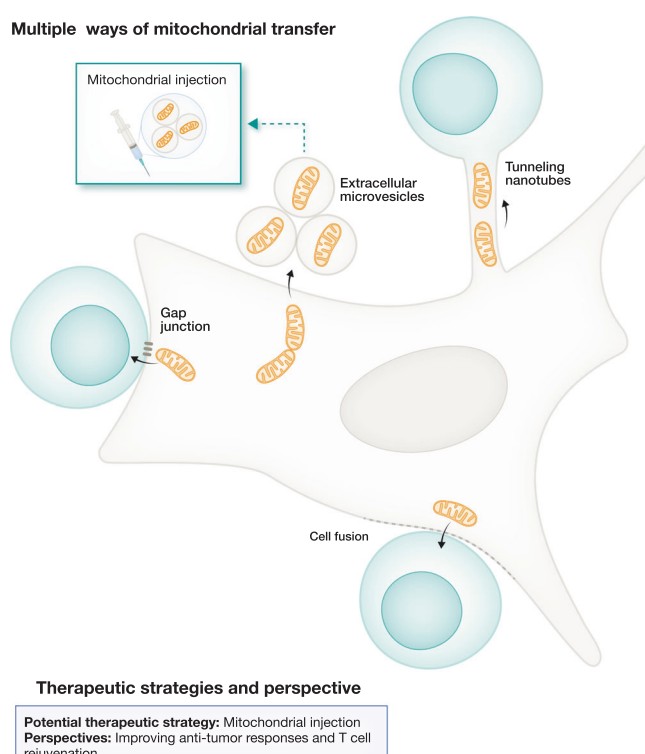

**Multiple ways of mitochondrial transfer**

Mitochondrial injection

Extracellular microvesicles

Tunneling nanotubes

Gap junction

Cell fusion

**Therapeutic strategies and perspective**

**Potential therapeutic strategy:** Mitochondrial injection
**Perspectives:** Improving anti-tumor responses and T cell rejuvenation

**Figure 5. Mitochondrial transfer and its therapeutic potentials.**

Mitochondria can be transferred between cells as another approach to give away mitochondria via multiple pathways, including gap junctions, cell fusion, extracellular vesicles and tunnelling nanotubes. T cells can act as both donors and recipients of mitochondria with such transfers, enhancing metabolic fitness, anti-tumour immunity and cellular rejuvenation. Green arrow with dashed lines indicates proposed therapeutic potential.

**Box 1   In need of answers**

- What molecular mechanisms are involved in the selective targeting of mitochondria for transfer or asymmetric partitioning?

- Is mitochondrial diversity within individual T cells linked to different functions, and how can we identify and target these distinct mitochondrial types?

- Is it enough to target dysfunctional mitochondria to rejuvenate T cells and organismal ageing, considering the detrimental ageing microenvironment?

- How can we convert the knowledge of mitochondrial quality control into translational medicine and improve current therapeutics?

We already know that mitochondrial ROS production increases with age, and given the inefficiency of autophagy with age, mitophagy is likely to decline in most cell types, and asymmetric cell division follows a similar trajectory. The efficiency of mitochondrial transfer remains low (less than 10%) in T cells, even in young organisms, whether this efficiency is further compromised during ageing requires additional investigation. Perhaps the mechanisms to remove mitochondria decline with age because these processes are energy-consuming. ATP becomes scarce with age, with cells relying primarily on glycolysis to meet their metabolic demands, and these processes may be down-regulated as mitochondria no longer need to be so functional.

Although the diminished function of mechanisms that promote mitochondrial recycling can result in higher levels of mitochondrial heterogeneity, with accumulation of mitochondria exhibiting mutated mtDNA, mitochondrial function becomes more homogeneous with ageing, as they exhibit lower functional activity. This loss of mitochondrial selection results in a more homogeneous pool of cells, reducing their adaptability to life-threatening environmental changes that occur during the ageing process, such as infection, cancer, and other stressors.

If we were able to modulate these processes genetically or pharmacologically, we would be in a position to determine what roles they play during ageing and other pathological processes like cancer and autoimmune diseases. Advances in precision gene-editing tools now allow for the integration of mitochondrial therapeutics with conventional treatments — for instance, enhancing the metabolic fitness of CAR-T cells (Van Bruggen et al, 2019). A recent breakthrough demonstrated targeted mtDNA-editing in human cells by co-delivering a DNA end-joining system and site-specific mitochondrial nucleases, opening new possibilities for direct mitochondrial genome engineering (Fu et al, 2025). Alternative strategies, such as selectively degrading mitochondrial proteins via Degron or PROTAC technologies, could further dissect the functional contributions of specific mitochondrial components and fine-tune mitochondrial activity. Additionally, improved tools are required to monitor mitochondrial function after therapeutic interventions in both preclinical and clinical settings. For example, the fate of transferred exogenous mitochondria varies depending on the donor cells, making them targets of mitophagy (Lin et al, 2024), or not (Ikeda et al, 2025) in the recipient cells. Given the heterogeneous environments in which immune cells can be found, there is also a big demand to study the single-cell metabolism of immune cells during ageing, especially in the

pronounced organismal ageing when manipulated to adopt an unhealthy state. One such example are mice with a deletion of the DNA repair factor (Ercc1) in hematopoietic cells only (Yousefzadeh et al, 2021). Another example are mice with endothelial cells lacking the vascular endothelial growth factor (Vegf) (Grunewald et al, 2021). Both mouse models develop a systemically aged phenotype.

Targeting mitochondrial function and degradation through mitophagy are undoubtedly crucial for preserving mitochondrial health. However, there are still significant questions that are awaiting answers (see Box 1). The in vivo and physiological significance of the other two mitochondrial removal mechanisms described in this review (ACD and transfer of mitochondria) is yet to be demonstrated. For both processes, we know little about the machinery that is involved. How does the cell recognise different mitochondrial signatures within the cytoplasm? How are specific mitochondria pulled to one side during asymmetric cell division or being transferred to another cell? How are the mitochondria selected and recognised by the respective machineries? How long does a pool of dysfunctional mitochondria need to be maintained to permanently change the fate of a T cell? What are the signalling molecules involved in this? Does it involve nucleo-mitochondrial communication and epigenetic changes?

context of tissue biology. Emerging spatial single-cell metabolomics techniques (Hu et al, 2023; Saunders et al, 2023) and high-throughput flow cytometry approaches like SCENITH now enable such investigations (Argüello et al, 2020). As these methodologies advance, we will unravel the intricate relationship between mitochondrial health and immune cell function in ageing, enabling the creation of context-dependent strategies to manipulate T cell function, rejuvenate T cells and delay organismal ageing.

## Peer review information

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

## Acknowledgements

This work was supported by Wellcome Trust 220784/Z/20/Z (AKS), Helmholtz Association REK-0157 (AKS), Chinese Academy of Medical Sciences (CAMS) Innovation Fund for Medical Sciences (CIFMS) 2018-I2M-2-002 (LL), Wellcome Trust 220452/Z/20/Z (MB) and La Caixa Foundation LCP/BQ/PI24/12040005 (AVL-V).

## Author contributions

**Lin Luo**: Conceptualisation; Funding acquisition; Visualisation; Writing—original draft; Writing—review and editing. **Ana Victoria Lechuga-Vieco**: Conceptualisation; Visualisation; Writing—original draft; Writing—review and editing. **Clara Sattentau**: Conceptualisation; Visualisation; Writing—review and editing. **Mariana Borsa**: Conceptualisation; Supervision; Visualisation; Writing—original draft; Writing—review and editing. **Anna Katharina Simon**: Supervision; Funding acquisition; Visualisation; Writing—original draft; Writing—review and editing.

## Funding

## Disclosure and competing interests statement

The authors declare no competing interests.

