## [Peer Review File · EMBO Reports]

Dysfunctional mitochondria in ageing T cells: a perspective on mitochondrial quality control mechanisms

Lin Luo, Ana Lechuga Vieco, Clara Sattentau, Mariana Borsa, and Anna Katharina Simon

Corresponding author(s): Anna Katharina Simon (katja.simon@mdc-berlin.de)

Review Timeline:

Submission Date:	20th Dec 24
Editorial Decision:	28th Jan 25
Revision Received:	30th Apr 25
Editorial Decision:	5th Jun 25
Revision Received:	10th Jun 25
Accepted:	16th Jul 25

Editor: Achim Breiling

Transaction Report:

Dear Dr. Simon,

Thank you for the submission of your review article to our editorial offices. I have now received the full set of referee reports that is copied below. As you will see, all three referees state that your manuscript is interesting and timely. However, they have several suggestions to improve the submission that I kindly ask you to address in a revised manuscript.

Given the constructive referee comments, I would thus like to invite you to revise your manuscript with the understanding that all referee points will be addressed in the revised manuscript and in a detailed point-by-point response.

I further have these editorial requests:

- Please add up to 5 keywords to the manuscript and place these below the abstract.
 - We have space for 1 more figure, and it would be nice to have indeed 4 figures, as we encourage authors to maximize the use of visual elements, which will increase the accessibility of the piece to a non-specialist readership. Please consider adding 1 more figure and note the instructions regarding figures below.
 - We usually ask our authors to include a box called "In need of answers" that briefly outlines the major questions that are still open in a given field in the form of a few bullet points. These questions can be accompanied by a brief explanation of what would be needed to address them and may provide helpful towards setting the stage for future experimentation in the field. For an example see this recent review we published: <https://www.embopress.org/doi/full/10.1038/s44319-024-00135-4>
 - Please also add callouts for the box to the manuscript text (Box 1).
 - Please also make sure the references and their callouts are formatted according to our reference format (with et al. for manuscripts with more than 10 authors):
<http://www.embopress.org/page/journal/14693178/authorguide#referencesformat>
 - We now use CRediT to specify the contributions of each author in the journal submission system. CRediT replaces the author contribution section. Please use the free text box to provide more detailed descriptions and do NOT provide your final manuscript text file with an author contributions section. See also our guide to authors:
<https://www.embopress.org/page/journal/14693178/authorguide#authorshippinguidelines>
 - Please make sure that all the funding information is also entered into the online submission system and that it is complete and similar to the one in the acknowledgement section of the manuscript text file.
- I think this is a very interesting review and while I appreciate that incorporating the referees' suggestions will still require some work, I am convinced that the article is worth it and will benefit from it.

When submitting your revised manuscript, we will require a Microsoft Word file (.doc) of the revised manuscript text including detailed figure legends (at the very end), but without the figures.

Please provide the final figures as separate, high-resolution files as .pdf, .eps, .tif, or .jpg (one file per figure). Please finalize the drafts provided and make sure they accurately illustrate the key scientific concepts that you wish to show.

Please also note the following points:

- If there are certain aspects of your figure draft that are based upon assumptions or where the scientific data remains ambiguous (for example, schematically depicting a presumed direct protein-protein interaction, protein shape or subcellular localizations etc.) please add a comment so that we can work with you on an accurate depiction. Please ensure the directionality and nature of interactions is presented accurately.
- If the figure or single panels of the figure have been adapted from a published figure, please add this information to the figure legend (e.g., 'Adapted from...' or 'Based on...'). The editor will discuss if a reference and permission will be necessary
- Please only re-use figures or parts of a figure if this is essential for understanding the concept communicated. Often a reference to a previous paper will suffice. If the figure contains re-used images or elements of images, including schematics, micrographs or photos, please make sure that you have the permission/license to publish it (this also applies to your own previous work, if the journal you published in retains copyright. Certain 'creative commons' open access licenses, such as CC-BY 4.0, allow re-use without additional formal permissions). All re-used material must be explicitly cited.

- If you use an image data base for scientific iconography (e.g., BioRender), please let us know if you have a license that allows for publication in an academic journal. Often authors use misleading iconography for expedience. Please ensure the information shown is scientifically accurate. If in doubt, please discuss with the editor or provide a sketch so that our designers can create accurate iconography.

- For figures created using a software for editing vector objects like Inkscape, CorelDraw etc., please send the file as a PDF (or SVG, or EPS), PowerPoint or Keynote in which the labels and objects are still editable. For figures created using Adobe Illustrator, please send the Illustrator (.ai) file.

I look forward to seeing a revised version of your manuscript when it is ready. Please let me know if you have questions or comments regarding the revision.

Kind regards,

Achim

Referee #1:

This paper is a review on mitochondrial dynamics in T cells. Specifically, it goes into how T cells optimize their function through adaptations in mitochondria, and how these processes presumably change with age. It reviews mechanisms of mitochondrial adaptation (e.g. mitophagy, ACD, mitochondrial transfer), as well as mechanisms of mitochondrial dysfunction (e.g. decreased mitochondrial biogenesis, increased ROS, etc.). Overall, this was a comprehensive review on a topic that is garnering lots of attention at the moment, particularly in the cancer field.

Please note that there may be multiple overlapping concepts with the recently published review (2023): Mitochondria during T cell aging (published in Seminars in Immunology).

Minor suggestions:

Intro -

Page 2, paragraph 2:

I would suggest adding a sentence in this paragraph to address the concept of how increased inflammation doesn't inherently equal increased resistance to infection/cancer. It's reasonable to think more inflammation=better at combating infections and cancer. Therefore, addressing that there is a dichotomy between the fact that there is inflammation but more susceptibility to infection/cancer may help the reader resolve a somewhat counterintuitive concept.

Page 2, paragraph 3:

The topic sentence of this paragraph should also include how T cells are central in combatting cancer, along with infections!

Page 2, paragraph 4:

First sentence of this paragraph states that metabolic adaptation is important for fueling metabolic demands - my opinion is it would be more appropriate to say functional demands?

Page 3, paragraph 5:

This sentence "Functional deterioration of T cells during ageing generates T cells, contributing to age-associated diseases either by an excessive inflammatory profile, or by decreased function, predisposing to infections, cancer progression or accumulation of senescent cells" needs revision or to be split into two sentences to improve clarity.

Mitochondrial metabolism during T cell ageing -

Page 4 and 5, paragraph 2 under "Mechanisms of Dysfunctional mitochondria":

I think this paragraph needs restructuring or to be split into multiple paragraphs, to facilitate clarity of the concepts (which are complex to someone who is not an expert in the field).

Specifically, this section "In addition, it has been shown that the SASP results from dysfunctional mitochondria, which lead to the downregulation of the NAD⁺/NADH ratio. This activates the AMPK signalling pathway, inducing the phosphorylation of p53 and the expression of p16 and p21, which contribute to mitotic arrest. Levels of NAD⁺ decline with age and is coupled with increased CD38 expression. CD38 is a NADase that is responsible for age-related mitochondrial dysfunction [33]" feels like a conglomeration of concepts that interrupt the flow of the paragraph, and to someone who isn't an expert in this particular topic (i.e. mechanisms of mitochondrial biogenesis), its hard to see how it fits in with the rest of the paragraph.

Additionally, there are some contradictory statements in the paragraph that need some additional explanation:

- 1) AMPK: The first part of the paragraph states that AMPK activation is "good" because it facilitates mito biogenesis. However, later in the paragraph you state that the activation of the AMPK signalling pathway contribute to mitotic arrest (which I'm unsure how that relates to mitochondria)
- 2) PGC1alpha: at the beginning of the paragraph, you talk about PGC1-alpha as being critical for stimulating mito biogenesis, which slows down with age. But later on you say that mTORC, which activation is associated with aging, stimulates PGC1-alpha. I would suggest adding a sentence to address these contradictory concepts.

Page 5, paragraph 1 under "Mitochondria-associated changes in T cells during ageing":

The sentence "Another signal downstream when naïve cells first encounter with an antigen, is signal transducer and activator of transcription 3 (STAT3)" read awkward, I am unsure if it is grammatically correct.

Page 5 and 6, paragraph 1 under "Mitochondria-associated changes in T cells during ageing":

You talk about differences in metabolite compositions in older and younger plasma after influenza vaccination - may be interesting to add a sentence talking about what these differences are? E.g. is there an abundance of immune suppressive metabolites in the aged population?

Page 8, paragraph 2 under "Drugs and interventions that target mitochondria products":

Some good papers that would fit in nicely in this paragraph (when you talk about mitochondria and anti-tumor capacity):

- 1) Jaco A. C. van Bruggen, Anne W. J. Martens, Joseph A. Fraietta, Tom Hoffland, Sanne H. Tonino, Eric Eldering, Mark-David Levin, Peter J. Siska, Sanne Endstra, Jeffrey C. Rathmell, Carl H. June, David L. Porter, J. Joseph Melenhorst, Arnon P. Kater, Gerritje J. W. van der Windt; Chronic lymphocytic leukemia cells impair mitochondrial fitness in CD8⁺ T cells and impede CAR T-cell efficacy. *Blood* 2019; 134 (1): 44-58. doi: <https://doi.org/10.1182/blood.2018885863>
- 2) Lontos K, Wang Y, Joshi SK, et al Metabolic reprogramming via an engineered PGC-1 α improves human chimeric antigen receptor T-cell therapy against solid tumor *Journal for ImmunoTherapy of Cancer* 2023;11:e006522. doi: 10.1136/jitc-2022-006522

Mitophagy to delay ageing -

Page 10, paragraph 1 under "Mitophagy impacts T cells and T cell ageing ":

In this paragraph, you discuss how knocking out autophagy genes may decrease CD8 T memory cells. There has been contradictory research on this topic that you may want to discuss, e.g. Devorkin et al. (2019) found that knocking out Atg5 led to an increase in effector memory cells.

Reference:

DeVorkin L, Pavey N, Carleton G, Comber A, Ho C, Lim J, McNamara E, Huang H, Kim P, Zacharias LG, Mizushima N, Saitoh T, Akira S, Beckham W, Lorzadeh A, Moksa M, Cao Q, Murthy A, Hirst M, DeBerardinis RJ, Lum JJ. Autophagy Regulation of Metabolism Is Required for CD8⁺ T Cell Anti-tumor Immunity. *Cell Rep.* 2019 Apr 9;27(2):502-513.e5. doi: 10.1016/j.celrep.2019.03.037. PMID: 30970253.

Page 10, paragraph 1 under "Mitophagy impacts T cells and T cell ageing ":

In some parts of the paper, you have the definition of the acronym first and then the acronym in brackets, while here you have the opposite (i.e. AICD (activation-induced cell death)). Just pointing out for consistency purposes.

Page 11, paragraph 1 under "Drugs and interventions that target mitophagy ":

The sentence "We treated old mice with spermidine and found that it can recover T cell memory responses, an effect that is not observed in the absence of autophagy", needs more explanation. How does spermidine relate/target mitophagy?

Asymmetric inheritance of mitochondria -

Page 11:

The heading and first subheading are both labelled "Asymmetric inheritance of mitochondria". I would suggest changing one of these headings so they aren't identical.

Mitochondria Transfer -

Page 16, paragraph 1 under "Mitochondrial transfer form and to T cells ":

A very recent publication shows mito transfer from cancer cells to T cells cause T cell dysfunction. Might fit in well with this section.

Reference:

Ikeda, H., Kawase, K., Nishi, T. et al. Immune evasion through mitochondrial transfer in the tumour microenvironment. Nature (2025). <https://doi.org/10.1038/s41586-024-08439-0>

Overall comments/suggestions on manuscript -

- I thought your manuscript covered the topic of mitochondria and T cells as it relates to aging very comprehensively. One section that I would have liked to see expanded on was the "perspectives" section. Specifically, it would be nice to hear specific, concrete ways on how you think improving mitochondrial health will be implemented into health (e.g. aging) and medicine (e.g. cancer, autoimmunity, etc.) in the future, and how feasible these ideas are with our current technology. The last paragraph of the manuscript does touch on this a little, but it would be great to see more in depth discussion on this.

Figures -

Figure 2, under panel "Asymmetric inheritance of mitochondria during mitosis"
What is the difference between the orange vs. grey mitochondria?

Referee #2:

Review summary:

The manuscript identifies four main mechanisms through which T cells manage mitochondrial damage: (1) metabolic adaptation, which regulates mitochondrial metabolism to meet cellular demands; (2) mitochondrial degradation, including mitophagy, to remove damaged mitochondria; (3) asymmetric mitochondrial inheritance during mitosis, which segregates damaged mitochondria to daughter cells; and (4) mitochondrial transfer, facilitating the exchange of healthy mitochondria between cells. For each mechanism, the review discusses the underlying processes, their impact on T cell ageing, and current or potential therapeutic interventions aimed at mitigating dysfunction. While the manuscript comprehensively addresses these pathways, there is some overlap and inconsistency in terminology, such as distinguishing mitochondrial removal from other mechanisms of damage management. Greater clarity and distinction between these mechanisms would strengthen the narrative.

Comments and suggested improvements:

1. Major claims, novelty and who will be interested

The manuscript highlights the role of mitochondrial in T cell ageing and its systemic impacts on ageing and immunosenescence. These claims are significant as they integrate mitochondrial biology with immune ageing. As such, the review will interest researchers in immunology, metabolism, ageing, and translational research. The review also focuses on therapeutic strategies making the review more innovative. The exploration of mitochondrial transfer and asymmetric inheritance adds novelty. However, the data supporting it are limited and should be expanded to be convincing.

2. Content imbalance

There is an imbalance in the content - there is not enough content on mitochondria role in T cell and organismal ageing but extensive details on current therapeutic strategies to remove dysfunctional mitochondria when it is then concluded that little is known about mitochondria roles in T cells and it may not reverse immune ageing.

3. Earlier literature

The manuscript synthesizes existing data and is well-referenced but occasionally misses some references within the text. It could benefit from more critical appraisal of contradictory findings such as models of mitophagy.

4. Clarity of writing

It seems like multiple people wrote it in different styles. It would be best to make it more consistent between each paragraphs throughout to improve the flow. For example, the mitophagy section is the only one where publications are mentioned as Author et al.

5. Title

The title falls a bit short and could be more appropriate for the current review as it focuses more on targeting mitochondrial

dysfunction compared to mitochondria as a key driver of T cell and organismal ageing.

6. Figures

Figure 1 needs more information and clarity. Perhaps a figure detailing how mitochondrial metabolism affects T cells. Figure 2 and 3 are similar and can be condensed into one figure. Figure 2 has a cell in the middle that is not needed and the therapeutic interventions from Figure 3 could be added in Figure 2

Referee #3:

The review by Luo et al provides a nice summary of mitochondria functions in regulating T cell and organismal aging. In general, the manuscript is well written and the knowledge of current literature on the topic is nicely conveyed. The discussion on discrepancies on some discoveries throughout the manuscript is enjoyable and will be beneficial for the readers. I only have minor comments:

1. A couple of statements need to be clarified:

"In addition, under nutrient-rich conditions, mammalian target of rapamycin complex 1 (mTORC1), which senses nutrients in the cell, becomes active and suppresses mitochondrial biogenesis [25]. Upregulated mTORC1 activity is one of the hallmarks of ageing [3]. It controls key downstream molecules including nuclear respiratory factor (NRF) 1 and 2 for mitochondrial gene transcription, PPARs for lipid metabolism, estrogen-related nuclear receptors (ERRs) for oxidative phosphorylation, and cyclic AMP-responsive element-binding protein 1 (CREB1) and FOXO to induce PGC-1 α transcription, collectively facilitating mitochondrial signalling." The first sentence says active mTORC1 suppresses mitochondrial biogenesis, but the latter sentences indicate that upregulation of mTORC1 activity can induce PGC-1 α transcription which increase mitochondrial biogenesis. Can the authors explain/discuss?

"Human CD8⁺ TEMRA cells are found to have reduced proliferative ability and undergo anaerobic glycolysis with higher ROS production and reduced mitochondrial membrane potential [44]. This can be enhanced through p38 inhibition which improves mitochondrial function without successfully modulating their metabolism, suggesting cell-intrinsic metabolic changes independent of p38 signalling." I find the word "This" here confusing because it is not clear what "This" refers to. Please revise the sentence.

2. Protein names such as cdkn, Tnf, Usp30 etc should be capitalized. Please go through the manuscript and correct the nomenclature for proteins and genes.

3. Citations should be improved throughout the manuscript. For example, there are no references for PINK1 and Parkin mitophagy on page 9 or on page 10, the references on "limited studies that directly evidence the presence of mitochondrial molecules or the entire organelles among the autophagosomal cargo in aged organisms or provide definitive evidence of reduced mitophagy in mammalian cells during ageing" should be included. The statement "In fact, the deletion of key autophagy genes can indirectly lead to changes in mitochondria, for example via increased mitochondrial biogenesis or preferential survival of cells that retain mitochondria and may not directly stem from deleterious degradation." is also not supported by any references.

4. PINK1 and Parkin are not mitophagy receptors. The primary mitophagy receptors for PINK1/Parkin-dependent mitophagy pathways are OPTN and NDP52 (PMCID: PMC5018156; PMC4210283; PMC4592482). Please correct the mistake in the text and figure.

Nuffield Department of Orthopaedics,
Rheumatology and Musculoskeletal Sciences

Professor AK Simon, PhD

28th April 2025

Re: Revision of Manuscript #EMBOR-2024-61048V1 Dysfunctional mitochondria as drivers of T cell ageing-a perspective on novel mitochondrial quality control mechanisms

Dear Editor and Reviewers,

We would like to thank the editors for the helpful suggestions and sincerely appreciate the reviewers' time and constructive feedback on our manuscript titled '**Dysfunctional mitochondria as drivers of T cell ageing-a perspective on novel mitochondrial quality control mechanisms**'.

We have carefully considered all comments and have revised the manuscript accordingly. Please find below a point-by-point response to the reviewers' comments. To highlight the changes that we have made, the paragraphs in the revised manuscript are numbered. The changes responding to reviewer's points are highlighted in yellow and the changes to improve the writing clarity are highlighted in grey. Additionally, to provide clarification, we have added a box for questions in need of answers, numbered as Paragraph 39 (P39) with callouts highlighted in P41 in the manuscript, as requested by the editor.

- **Referee #1:**

This paper is a review on mitochondrial dynamics in T cells. Specifically, it goes into how T cells optimize their function through adaptations in mitochondria, and how these processes presumably change with age. It reviews mechanisms of mitochondrial adaptation (e.g. mitophagy, ACD, mitochondrial transfer), as well as mechanisms of mitochondrial dysfunction (e.g. decreased mitochondrial biogenesis, increased ROS, etc.). Overall, this was a comprehensive review on a topic that is garnering lots of attention at the moment, particularly in the cancer field.

Please note that there may be multiple overlapping concepts with the recently published review (2023): Mitochondria during T cell aging (published in Seminars in Immunology).

Reply to reviewer's point: María Mittelbrunn's review primarily focuses on the different mechanisms underlying mitochondrial dysfunction in T cells and its impact on their cell function, with a particular focus on mitochondrial metabolism and molecular mechanisms. In contrast, our review delves into cell biology and examines the mechanisms by which T cells cope with dysfunctional mitochondria, specifically by removing their byproducts, and selectively removing or degrading dysfunctional mitochondria. Here we provide novel perspectives on pathways that remain relatively unexplored and, in some cases, controversial in the context of T cells – namely, asymmetric cell division and mitochondrial transfer between cells. We also highlight the therapeutic potential of targeting these cellular processes to improve mitochondrial health and T-cell immunity.

We also specified this in the manuscript (P7): 'In a recent review, the molecular mechanisms underlying dysfunctional mitochondria during T cell ageing have been summarized, including alterations in mitochondrial DNA (mtDNA) genome stability, mitochondrial dynamics, Ca²⁺ homeostasis, mitochondrial biogenesis and degradation and oxidative stress [29]. Here, we will first discuss the role of mitochondrial ROS and mitochondrial biogenesis during T cell ageing.'

1. Minor suggestions:

Intro –

Page 2, paragraph 2:

I would suggest adding a sentence in this paragraph to address the concept of how increased inflammation doesn't inherently equal increased resistance to infection/cancer. It's reasonable to think more inflammation=better at combating infections and cancer. Therefore, addressing that there is a dichotomy between the fact that there is inflammaging but more susceptibility to infection/cancer may help the reader resolve a somewhat counterintuitive concept.

Reply to reviewer's point: We would like to thank the reviewer for the suggestion and have added the relevant information to the manuscript (P2) as follows: 'Counterintuitively, inflammaging challenges the limited capacity of an aged immune system to resolve

infection. For example, cytomegalovirus (CMV) infection, typically asymptomatic, becomes symptomatic in older women due to high levels of inflammatory cytokines that activate the replication of cytomegalovirus [9]. The release of pro-inflammatory cytokines also supports tumour growth and metastasis, resulting in higher susceptibility to cancers [10], [11].'

Page 2, paragraph 3:

The topic sentence of this paragraph should also include how T cells are central in combatting cancer, along with infections!

Reply to reviewer's point: In response to the reviewer's suggestion, we have included a description of how T cells combat cancers in the revised manuscript (P3), as follows: 'Cytotoxic CD8+ T cells recognize antigens presented by major histocompatibility complex (MHC) molecules I on the surface of infected or cancerous cells, which are then targeted for killing through effector molecules. CD4+ T helper cells recognise antigen through MHC class II and support the activation and persistence of CD8+ T cells, enhance antigen presentation by dendritic cells, and recruit other immune effector cells to the infection site or the tumour microenvironment. This coordinated response is also a fundamental component of the immune system's cancer surveillance mechanisms, although tumour antigens are more difficult to target due to their immune evasion and suppression potentials [13], [14].'

Page 2, paragraph 4:

First sentence of this paragraph states that metabolic adaptation is important for fueling metabolic demands - my opinion is it would be more appropriate to say functional demands?

Reply to reviewer's point: We would like to thank the reviewer for this comment. Taking into consideration both the energy and functional demands of cells, we have revised the corresponding section in the manuscript (P4) as follows: 'Existing evidence substantiates the role of metabolic adaptation during T cell activation, survival, and differentiation to fuel their functional demands [15], [16].'

Page 3, paragraph 5:

This sentence "Functional deterioration of T cells during ageing generates T cells, contributing to age-associated diseases either by an excessive inflammatory profile, or by decreased function, predisposing to infections, cancer progression or accumulation of senescent cells" needs revision or to be split into two sentences to improve clarity.

Reply to reviewer's point: We apologise for the confusion and thank the reviewer for pointing this out. We have rephrased the sentence (P5) as follows: 'These metabolic changes are closely related to mitochondrial dysfunction, which disrupt their multi-faceted role in maintaining T cell homeostasis.'

'Notably, CD8+ T cells show greater age-related decline than CD4+ T cells, with significant depletion of the naïve compartment due to severe mitochondrial dysfunction. These defects lead to reduced metabolic flexibility, impaired electron transport, and elevated ROS,

propagating oxidative stress across cellular compartments. The elevated ROS production and oxidative stress by damaged mitochondria hinder T cell differentiation and activation [27], [28].'

Mitochondrial metabolism during T cell ageing –

Page 4 and 5, paragraph 2 under "Mechanisms of Dysfunctional mitochondria":

I think this paragraph needs restructuring or to be split into multiple paragraphs, to facilitate clarity of the concepts (which are complex to someone who is not an expert in the field).

Reply to reviewer's point: Following the reviewer's suggestion, and considering that other reviews have already summarised the mechanisms of dysfunctional mitochondria during T cell ageing, we have refined this section by placing greater emphasis on mitochondrial ROS, biogenesis and translation. To enhance clarity and focus, the content has been reorganised into three paragraphs (P7-9) and several sentences (highlighted in grey) have been rephrased accordingly.

Specifically, this section "In addition, it has been shown that the SASP results from dysfunctional mitochondria, which lead to the downregulation of the NAD⁺/NADH ratio. This activates the AMPK signalling pathway, inducing the phosphorylation of p53 and the expression of p16 and p21, which contribute to mitotic arrest. Levels of NAD⁺ decline with age and is coupled with increased CD38 expression. CD38 is a NADase that is responsible for age-related mitochondrial dysfunction [33]" feels like a conglomeration of concepts that interrupt the flow of the paragraph, and to someone who isn't an expert in this particular topic (i.e. mechanisms of mitochondrial biogenesis), its hard to see how it fits in with the rest of the paragraph.

Reply to reviewer's point: We agree with the reviewer that this sentence is not relevant to the overall content, and have therefore removed it from the text. 'Levels of NAD⁺ decline with age and is coupled with increased CD38 expression. CD38 is a NADase that is responsible for age-related mitochondrial dysfunction [33]'.

Additionally, there are some contradictory statements in the paragraph that need some additional explanation:

1) AMPK: The first part of the paragraph states that AMPK activation is "good" because it facilitates mito biogenesis. However, later in the paragraph you state that the activation of the AMPK signalling pathway contribute to mitotic arrest (which I'm unsure how that relates to mitochondria)

Reply to reviewer's point: We would like to thank the reviewer for this comment and explain the correlation between energy scarce condition and mitotic arrest in the manuscript (P9) as follows: 'Thus, by lacking SIRT1 activity, ageing results in a scenario where AMPK is activated due to energy-scarce conditions, but this results in mitotic arrest – through phosphorylation of p53 and increasing p16 expression [39]. As one of the major features of cellular senescence, mitotic arrest indirectly helps enhance mitochondrial

biogenesis: When cells are arrested in mitosis, they shift focus from cell division to repairing damaged organelles helping to restore mitochondrial function. However, extended mitotic arrest can lead to a decline in mitochondrial membrane potential, triggering both mitophagy and cell death in senescent cells [40], [41]. This reflects the context-dependent regulation of mitochondrial biogenesis, where nutrient-rich and nutrient-deprived conditions trigger distinct signalling pathways with varying effects.'

2) PGC1alpha: at the beginning of the paragraph, you talk about PGC1-alpha as being critical for stimulating mito biogenesis, which slows down with age. But later on you say that mTORC, which activation is associated with aging, stimulates PGC1-alpha. I would suggest adding a sentence to address these contradictory concepts.

Reply to reviewer's point: We agree with the reviewer's point, and have expanded on the dual aspects of mitochondrial biogenesis (P9) as follows:

'Under conditions of energy deprivation, energy and nutrient sensing signals such as AMP-activated protein kinase (AMPK) and Sirtuin 1 (SIRT1) are activated, which both directly trigger PGC-1 α function by its phosphorylation and deacetylation. When PGC-1 α is activated, mitochondrial transcription factor A (TFAM) helps transporting both SIRT1 and PGC-1 α into the mitochondria to regulate mtDNA replication and transcription. SIRT1 is a NAD⁺ dependent protein deacetylase. As NAD⁺ levels decline during ageing, the resulting loss of NAD⁺ impairs SIRT1 activity, which in turn reduces the deacetylation of key regulators such as AMPK, forkhead box O (FOXO) proteins, and PGC-1 α .'

'Thus, by lacking SIRT1 activity, ageing results in a scenario where AMPK is activated due to energy-scarce conditions, but this results in mitotic arrest – through phosphorylation of p53 and increasing p16 expression [39]. As one of the major features of cellular senescence, mitotic arrest indirectly helps enhance mitochondrial biogenesis: When cells are arrested in mitosis, they shift focus from cell division to repairing damaged organelles helping to restore mitochondrial function. However, extended mitotic arrest can lead to a decline in mitochondrial membrane potential, triggering both mitophagy and cell death in senescent cells [40], [41]. This reflects the context-dependent regulation of mitochondrial biogenesis, where nutrient-rich and nutrient-deprived conditions trigger distinct signalling pathways with varying effects.'

Page 5, paragraph 1 under "Mitochondria-associated changes in T cells during ageing":

The sentence "Another signal downstream when naïve cells first encounter with an antigen, is signal transducer and activator of transcription 3 (STAT3)" read awkward, I am unsure if it is grammatically correct.

Reply to reviewer's point: We apologise for this confusing sentence. The sentence in question was intended to introduce an additional molecular mechanism by which ageing affects mitochondrial health in naïve T cells. However, as it is not clearly connected to the surrounding content, we have decided to remove this sentence to improve the clarity in this paragraph. We have rephrased this section in the manuscript (P10) as follows:

'In contrast, the activation of naïve CD8+ T cells has been shown to rely on mitochondrial biogenesis during early activation to generate effector cells and cytokines, demonstrated using both pharmacological and genetic tools that inhibit mitochondrial translation [45], [46]. Although evidence of the changes in mitochondrial translation in naïve T cells during ageing is lacking, increased mitochondrial oxidative stress and decreased mitochondrial membrane potential imply their dysfunctional status [45]. In fact, suboptimal proliferation and the development of apoptosis-prone phenotype in naïve CD8+T cells in the context of ageing have already been reported by others. These changes are associated with excessive fatty acid uptake and lipid storage, resulting in dysfunctional fatty acid oxidation during the activation of naïve T cells. The molecular drivers and correlation to mitochondrial dysfunction remain unclear [47].'

Page 5 and 6, paragraph 1 under "Mitochondria-associated changes in T cells during ageing":

You talk about differences in metabolite compositions in older and younger plasma after influenza vaccination - may be interesting to add a sentence talking about what these differences are? E.g. is there an abundance of immune suppressive metabolites in the aged population?

Reply to reviewer's point: We agree with the reviewer and have expanded the details of metabolite differences in the manuscript (P11) as follows:

'Metabolomic profiling of plasma from young and older healthy donors revealed distinct baseline metabolic signatures in response to influenza vaccination. Notably, young high responders (HRs) exhibited increased levels of upstream tryptophan metabolites—including L-tryptophan, 5-hydroxy-L-tryptophan (5-HTP), and indoleacetic acid—at 28 days post-vaccination, a pattern absent in older subjects. This observation aligns with established evidence that shunting of tryptophan metabolism suppresses T-cell responses [49]. The preservation of immunomodulatory upstream tryptophan metabolites in young HRs suggests maintained tryptophan bioavailability, potentially supporting T-cell responses driven by antigen presentation and interferon signalling. In contrast, older adults exhibit elevated triacylglycerols (TAGs) consumption, which are related to T memory and regulatory T cell (Treg) responses, suggesting older adults may rely on more T memory and Treg responses during influenza vaccination [50].'

Page 8, paragraph 2 under "Drugs and interventions that target mitochondria products":

Some good papers that would fit in nicely in this paragraph (when you talk about mitochondria and anti-tumor capacity):

1) Jaco A. C. van Bruggen, Anne W. J. Martens, Joseph A. Fraietta, Tom Hofland, Sanne H. Tonino, Eric Eldering, Mark-David Levin, Peter J. Siska, Sanne Endstra, Jeffrey C. Rathmell, Carl H. June, David L. Porter, J. Joseph Melenhorst, Arnon P. Kater, Gerritje J. W. van der Windt; Chronic lymphocytic leukemia cells impair mitochondrial fitness in CD8+ T cells and

impede CAR T-cell efficacy. Blood 2019; 134 (1): 44-58. doi: <https://doi.org/10.1182/blood.2018885863>

2) Lontos K, Wang Y, Joshi SK, et al Metabolic reprogramming via an engineered PGC-1 α improves human chimeric antigen receptor T-cell therapy against solid tumor Journal for ImmunoTherapy of Cancer 2023;11:e006522. doi: 10.1136/jitc-2022-006522

Reply to reviewer's point: We thank the reviewer for pointing out these two interesting papers which are highly relevant the topic of mitochondrial fitness and T cell function. We have added the relevant information to the manuscript (P15) as follows:

'It is also noteworthy that metabolic dysfunction in CD8+ T cells, driven by the immunosuppressive microenvironment of chronic lymphocytic leukaemia (CLL), contributes to CAR-T therapy resistance [72]. By co-expressing an inhibition-resistant PGC-1 α variant with anti-EGFR CARs, T cell metabolic fitness and tumour control are enhanced [73], revealing a promising strategy to synergise metabolic reprogramming with adoptive cell therapy (Figure 2).' These two papers are referenced as number 72-73.

Mitophagy to delay ageing –

Page 10, paragraph 1 under "Mitophagy impacts T cells and T cell ageing ":

In this paragraph, you discuss how knocking out autophagy genes may decrease CD8 T memory cells. There has been contradictory research on this topic that you may want to discuss, e.g. Devorkin et al. (2019) found that knocking out Atg5 led to an increase in effector memory cells.

Reply to reviewer's point: We appreciate the reviewer's suggestion of this paper. However, we would like to highlight that it provides limited evidence regarding the differentiation of Atg5-deficient cells into effector-memory cells, as the markers used to identify this subset are shared by short-lived effector CD8 T cells (CD44+ CD62L-). Furthermore, the manuscript does not directly address the role of mitophagy in regulating cell differentiation and effector anti-tumoral effector functions. We have nevertheless discussed this in the manuscript (P21) as follows:

'Loss of Atg5 has also been reported to promote increased CD8+ T cell anti-tumour activity, which authors suggest might be caused by their biased differentiation towards an effector-memory phenotype. Although evidence of enhanced effector functions in Atg5-deficient cells is compelling, which is accompanied by increased glucose metabolism and epigenetic regulation of effector and metabolic target genes, the role of autophagy and mitophagy was not directly addressed, neither was the distinction of short-lived effector cells and effector-memory T cells. Indeed, this work provides further evidence on the key role of autophagy in the long-term maintenance of stem-like quiescent T cells [91).'

Reference:

DeVorkin L, Pavey N, Carleton G, Comber A, Ho C, Lim J, McNamara E, Huang H, Kim P, Zacharias LG, Mizushima N, Saitoh T, Akira S, Beckham W, Lorzadeh A, Moksa M, Cao Q, Murthy A, Hirst M, DeBerardinis RJ, Lum JJ. Autophagy Regulation of Metabolism Is

Required for CD8+ T Cell Anti-tumor Immunity. Cell Rep. 2019 Apr 9;27(2):502-513.e5. doi: 10.1016/j.celrep.2019.03.037. PMID: 30970253.

Page 10, paragraph 1 under "Mitophagy impacts T cells and T cell ageing ":

In some parts of the paper, you have the definition of the acronym first and then the acronym in brackets, while here you have the opposite (i.e. AICD (activation-induced cell death)). Just pointing out for consistency purposes.

Reply to reviewer's point: We thank the reviewer for this comment and have amended the manuscript (P21) as follows: 'Activation-induced cell death (AICD) can only go ahead when general macroautophagy is inhibited [93].'

Page 11, paragraph 1 under "Drugs and interventions that target mitophagy ":

The sentence "We treated old mice with spermidine and found that it can recover T cell memory responses, an effect that is not observed in the absence of autophagy", needs more explanation. How does spermidine relate/target mitophagy?

Reply to reviewer's point: To answer the reviewer's question, we have included relevant information to the manuscript (P24) as follows:

'We treated old mice with spermidine and found that it can recover T cell memory responses, an effect that is not observed in the absence of autophagy [89]. Together with the studies showing a surplus of mitochondria in Atg7 knockout CD8+ T cells [89], this suggests that spermidine may have an effect on mitophagy. Alternatively, spermidine may turn on mitochondrial translation, as shown in macrophages [103].'

Asymmetric inheritance of mitochondria –

Page 11:

The heading and first subheading are both labelled "Asymmetric inheritance of mitochondria". I would suggest changing one of these headings so they aren't identical.

Reply to reviewer's point: We thank the reviewer for pointing this out and have modified the name as follows: 'Mitochondria as drivers of asymmetric fates'

Mitochondria Transfer –

Page 16, paragraph 1 under "Mitochondrial transfer from and to T cells ":

A very recent publication shows mito transfer from cancer cells to T cells cause T cell dysfunction. Might fit in well with this section.

Reference:

Ikeda, H., Kawase, K., Nishi, T. et al. Immune evasion through mitochondrial transfer in the tumour microenvironment. Nature (2025). <https://doi.org/10.1038/s41586-024-08439-0>

Kennedy Institute of Rheumatology | University of Oxford | Roosevelt Drive | Headington | Oxford | OX3 7FY
Email: katja.simon@kennedy.ox.ac.uk | Tel: +44 (0)1865 612627

Reply to reviewer's point: This paper is a timely and valuable addition, and we now discuss its findings in the manuscript (P38) as follows:

'Additionally, cancer cells in the tumour microenvironment transfer mutated mitochondria to tumour-infiltrating lymphocytes (TILs), impairing their immune response by inhibiting mitophagy and inducing metabolic impairment and senescence [169].'

Overall comments/suggestions on manuscript –

- I thought your manuscript covered the topic of mitochondria and T cells as it relates to aging very comprehensively. One section that I would have liked to see expanded on was the "perspectives" section. Specifically, it would be nice to hear specific, concrete ways on how you think improving mitochondrial health will be implemented into health (e.g. aging) and medicine (e.g. cancer, autoimmunity, etc.) in the future, and how feasible these ideas are with our current technology. The last paragraph of the manuscript does touch on this a little, but it would be great to see more in depth discussion on this.

Reply to reviewer's point: We thank the reviewer for the thoughtful feedback on our manuscript and appreciate the suggestion to expand the perspectives section. While the review focuses on ageing, we have incorporated a brief discussion on the role of rejuvenated T cells in cancer and autoimmunity. The relevant information has been added to the manuscript (P44) as follows:

'A recent breakthrough demonstrated targeted mtDNA-editing in human cells by co-delivering a DNA end-joining system and site-specific mitochondrial nucleases, opening new possibilities for direct mitochondrial genome engineering [171]. Alternative strategies, such as selectively degrading mitochondrial proteins via Degron or PROTAC technologies, could further dissect the functional contributions of specific mitochondrial components and fine-tune mitochondrial activity. Additionally, improved tools are required to monitor mitochondrial function after therapeutic interventions in both preclinical and clinical settings. For example, the fate of transferred exogenous mitochondria varies depending on the donor cells, making them targets of mitophagy [172], or not [169] in the recipient cells. Given the heterogeneous environments in which immune cells can be found, there is also a big demand to study single-cell metabolism of immune cells during ageing, especially in the context of tissue biology. Emerging spatial single-cell metabolomics techniques [173], [174] and high-throughput flow cytometry approaches like SCENITH now enable such investigations [175].'

Figures -

Figure 2, under panel "Asymmetric inheritance of mitochondria during mitosis"

What is the difference between the orange vs. grey mitochondria?

Reply to reviewer's point: We have revised the figures and added annotations to Figure 4. Figures are draft figures, to be drawn by professional designers at EMBO Reports. Figure 4 is as shown below:

Figure 4: Asymmetric Inheritance of Mitochondria and its therapeutic potentials. Following activation by antigen-presenting cells (APCs), T cells undergo mitosis. During this process, mitochondria can be inherited through asymmetric cell division (ACD). Selective mitochondrial inheritance through ACD offers a potential mechanism for clearing damaged mitochondria. Transient rapamycin treatment enhances ACD efficiency. Improving ACD also holds promise for improving CAR-T cell therapy and vaccination response. The figure was created with Biorender.com.

- **Referee #2:**

Review summary:

The manuscript identifies four main mechanisms through which T cells manage mitochondrial damage: (1) metabolic adaptation, which regulates mitochondrial metabolism to meet cellular demands; (2) mitochondrial degradation, including mitophagy, to remove damaged mitochondria; (3) asymmetric mitochondrial inheritance during mitosis, which segregates damaged mitochondria to daughter cells; and (4) mitochondrial transfer, facilitating the exchange of healthy mitochondria between cells. For each mechanism, the review discusses the underlying processes, their impact on T cell ageing, and current or potential therapeutic interventions aimed at mitigating dysfunction. While the manuscript comprehensively addresses these pathways, there is some overlap and inconsistency in terminology, such as distinguishing mitochondrial removal from other mechanisms of damage management. Greater clarity and distinction between these mechanisms would strengthen the narrative.

Comments and suggested improvements:

1. Major claims, novelty and who will be interested

The manuscript highlights the role of mitochondrial in T cell ageing and its systemic impacts on ageing and immunosenescence. These claims are significant as they integrate mitochondrial biology with immune ageing. As such, the review will interest researchers in immunology, metabolism, ageing, and translational research. The review also focuses on therapeutic strategies making the review more innovative. The exploration of mitochondrial transfer and asymmetric inheritance adds novelty. However, the data supporting it are limited and should be expanded to be convincing.

Reply to reviewer's point: We agree that current data supporting asymmetric inheritance and mitochondrial transfer in T cells remains limited. However, this scarcity highlights the need for further research, particularly within T-cell biology. Recent findings in the cell biology and immunology fields have uncovered multiple mechanisms of mitochondrial removal or delivery in different cell types, including mitochondrial transfer, asymmetric cell division, and mito-extrusion. Here, we discussed what has been shown experimentally in T cells: mitochondrial transfer and asymmetric cell division of T cell mitochondria, both of which represent novel strategies by which T cells manage mitochondrial quality to support their function. We understand this is still an emerging field which calls for more experimental exploration. However, in this review, we have commented on all the papers that involve these two mechanisms in T cells.

2. Content imbalance

There is an imbalance in the content - there is not enough content on mitochondria role in T cell and organismal ageing but extensive details on current therapeutic strategies to remove dysfunctional mitochondria when it is then concluded that little is known about mitochondria roles in T cells and it may not reverse immune ageing.

Reply to reviewer's point: This was our intention as previous reviews already exist on mitochondria in T cells (see reviewer 1). We have now mentioned this in the review's introduction. As for their role in organismal ageing, we have cited the two papers that exist on this topic. To reflect this emphasis, we have now changed the title of this review to 'Dysfunctional mitochondria as drivers of T cell ageing- a perspective on novel mitochondrial quality control mechanisms'

:

3. Earlier literature

The manuscript synthesizes existing data and is well-referenced but occasionally misses some references within the text. It could benefit from more critical appraisal of contradictory findings such as models of mitophagy.

Reply to reviewer's point: We are grateful for the reviewer's comments. As for the general criticism, we have tried to do exactly this in this section and are unsure how to cite the literature in a more critical way. We have added relevant information to the context of mitophagy (P20) as follows:

'However, studies using mammalian cells during ageing show contradictory results [82]. Indeed, a recent study reported no obvious decline in mitophagy in mice across all brain tissues with age, but rather region-specific dynamics [83]. Another study reports that while autophagy levels decline, mitophagy levels remain stable in several tissues analysed [84]. This field requires more evidence from other tissues and better quantitative in vivo tools. For instance, the detection of mitochondrial proteins in the autophagic cargo of aged cells would provide direct evidence of mitophagy's role in healthy ageing which is still lacking despite the development of novel tools to identify autophagosomal cargo [85], [86].'

4. Clarity of writing

It seems like multiple people wrote it in different styles. It would be best to make it more consistent between each paragraph throughout to improve the flow. For example, the mitophagy section is the only one where publications are mentioned as Author et al.

Reply to reviewer's point: Following the reviewer's suggestion, the writing styles have been harmonised and the writing clarity has been enhanced throughout the revised manuscript and are highlighted in grey throughout the manuscript.

5. Title

The title falls a bit short and could be more appropriate for the current review as it focuses more on targeting mitochondrial dysfunction compared to mitochondria as a key driver of T cell and organismal ageing.

Reply to reviewer's point: We thank the reviewer's comment, and have changed the title of this review to reflect relevant information in the manuscript(P2) as follows:

'Dysfunctional mitochondria as drivers of T cell ageing- a perspective on novel mitochondrial quality control mechanisms'

6. Figures

Figure 1 needs more information and clarity. Perhaps a figure detailing how mitochondrial metabolism affects T cells. Figure 2 and 3 are similar and can be condensed into one figure. Figure 2 has a cell in the middle that is not needed and the therapeutic interventions from Figure 3 could be added in Figure 2

Reply to reviewer's point: We have revised the figures, please find attached. Figures are draft figures, to be redrawn by professional figure designers at EMBO Reports.

Referee #3:

The review by Luo et al provides a nice summary of mitochondria functions in regulating T cell and organismal aging. In general, the manuscript is well written and the knowledge of current literature on the topic is nicely conveyed. The discussion on discrepancies on some discoveries throughout the manuscript is enjoyable and will be beneficial for the readers. I only have minor comments:

1. A couple of statements need to be clarified:

"In addition, under nutrient-rich conditions, mammalian target of rapamycin complex 1 (mTORC1), which senses nutrients in the cell, becomes active and suppresses mitochondrial biogenesis [25]. Upregulated mTORC1 activity is one of the hallmarks of ageing [3]. It controls key downstream molecules including nuclear respiratory factor (NRF) 1 and 2 for mitochondrial gene transcription, PPARs for lipid metabolism, estrogen-related nuclear receptors (ERRs) for oxidative phosphorylation, and cyclic AMP-responsive element-binding protein 1 (CREB1) and FOXO to induce PGC-1 α transcription, collectively facilitating mitochondrial signalling." The first sentence says active mTORC1 suppresses mitochondrial biogenesis, but the latter sentences indicate that upregulation of mTORC1 activity can induce PGC-1 α transcription which increase mitochondrial biogenesis. Can the authors explain/discuss?

Reply to reviewer's point: We thank the reviewer for noticing this contradiction, and have discussed this in the manuscript (P9) as follows:

'Under conditions of energy deprivation, energy and nutrient sensing signals such as AMP-activated protein kinase (AMPK) and Sirtuin 1 (SIRT1) are activated, which both directly trigger PGC-1 α function by its phosphorylation and deacetylation. When PGC-1 α is activated, mitochondrial transcription factor A (TFAM) helps transporting both SIRT1 and PGC-1 α into the mitochondria to regulate mtDNA replication and transcription. SIRT1 is a NAD⁺ dependent protein deacetylase. As NAD⁺ levels decline during ageing, the resulting loss of NAD⁺ impairs SIRT1 activity, which in turn reduces the deacetylation of key regulators such as AMPK, forkhead box O (FOXO) proteins, and PGC-1 α .'

"Human CD8⁺ TEMRA cells are found to have reduced proliferative ability and undergo anaerobic glycolysis with higher ROS production and reduced mitochondrial membrane potential [44]. This can be enhanced through p38 inhibition which improves mitochondrial function without successfully modulating their metabolism, suggesting cell-intrinsic metabolic changes independent of p38 signalling." I find the word "This" here confusing because it is not clear what "This" refers to.

Reply to reviewer's point: In response to the reviewer's comment, while p38 signalling is related to TEMRA senescent phenotypes, yet the underlying mechanisms remain unclear. How p38 signalling impacts mitochondrial health is indirectly described. In this case, we have revised the

text to avoid confusion and redundancy (P13): ‘Higher ROS production and reduced mitochondrial membrane potential are observed in TEMRAs compared to other subsets of CD8+ T cells [57].’

2. Protein names such as cdkn, Tnf, Usp30 etc should be capitalized. Please go through the manuscript and correct the nomenclature for proteins and genes.

Reply to reviewer’s point: We have harmonised the gene names and protein names throughout the text. These are highlighted in yellow for reference.

3. Citations should be improved throughout the manuscript. For example, there are no references for PINK1 and Parkin mitophagy on page 9 or on page 10, the references on "limited studies that directly evidence the presence of mitochondrial molecules or the entire organelles among the autophagosomal cargo in aged organisms or provide definitive evidence of reduced mitophagy in mammalian cells during ageing" should be included. The statement "In fact, the deletion of key autophagy genes can indirectly lead to changes in mitochondria, for example via increased mitochondrial biogenesis or preferential survival of cells that retain mitochondria and may not directly stem from deleterious degradation." is also not supported by any references.

Reply to reviewer’s point: We thank the reviewer for the suggestion. As this does not draw from existing studies, we made it clear this is a speculation in the manuscript as follows (P21): ‘Aside from this study, a general drawback of early research was the initial challenge of studying mitophagy isolated from general autophagy. One could speculate that the deletion of key autophagy genes could indirectly lead to changes in mitochondria, for example via increased mitochondrial biogenesis or preferential survival of cells that retain mitochondria and may not directly stem from deleterious degradation.’

4. PINK1 and Parkin are not mitophagy receptors. The primary mitophagy receptors for PINK1/Parkin-dependent mitophagy pathways are OPTN and NDP52 (PMCID: PMC5018156; PMC4210283; PMC4592482). Please correct the mistake in the text and figure.

Reply to reviewer’s point: Following the reviewer’s suggestion, we have corrected this in the manuscript(P19) as follows: ‘The PINK1/Parkin pathway is found not to be functional in human peripheral mononuclear cells of which T lymphocytes are one subset. The mitophagy receptors used in the PINK1/Parkin pathway are NDP52, OPTN and TAX1BP1 [78].’

katia

Dear Dr. Simon

Thank you for the submission of your revised review manuscript to our offices. I have now received the enclosed reports from the referees that were asked to re- assess it. As you will see, the referees now fully support publication of the manuscript.

Referee #2 has some suggestions to improve the figures, I ask you to address in a final revised manuscript. Moreover, please provide the figures without their headlines/titles within the figures themselves.

All the best,

Referee #1:

The authors have adequately addressed all of my concerns and queries. The review article, in my opinion, will be a nice piece for a broad audience.

Referee #2:

We thank the authors for addressing our previous comments and for the improvements made to the manuscript. The revised figures, particularly the separation of distinct mitochondrial dysfunctions, enhance clarity and allow for more focused interpretation.

To further strengthen the visual impact and utility of the figures, we recommend indicating where key therapeutic strategies are proposed to act. This would provide direct insight into the mechanisms of intervention and move the figures beyond a list-like summary toward more integrative illustrations. For example:

- In Figure 2, it would be helpful to indicate the points of action for Coenzyme A and NAD⁺ precursors.
- In Figure 3, including urolithin A with an arrow pointing toward PINK1 would clarify its role in modulating mitophagy.
- In Figure 4, we suggest briefly explaining the terms "proximal" and "distal" in the figure legend, in addition to the main text, to support reader understanding.
- In Figure 5, we recommend reintroducing the microvesicles injection with mitochondria as in the previous manuscript.

These additions would help highlight relevant therapeutic avenues.

Referee #3:

The authors have adequately addressed my comments and the review can now be published.

Nuffield Department of Orthopaedics,
Rheumatology and Musculoskeletal Sciences

Professor AK Simon, PhD

9th June 2025

Re: Revision of Manuscript #EMBOR-2024-61048V2 Dysfunctional mitochondria as drivers of T cell ageing-a perspective on novel mitochondrial quality control mechanisms

Dear Editors and Reviewers,

We would like to thank the editors for the helpful suggestions and sincerely appreciate the reviewers' time and valuable feedback on our manuscript titled '**Dysfunctional mitochondria as drivers of T cell ageing-a perspective on novel mitochondrial quality control mechanisms**'.

We have carefully considered all the comments and have revised the manuscript accordingly. Please find below a point-by-point response to the reviewers' comments.

- **Referee #1:**

The authors have adequately addressed all of my concerns and queries. The review article, in my opinion, will be a nice piece for a broad audience.

Reply to reviewer's point: We thank the reviewer for the positive comments. We hope that this review serves as a valuable and accessible summary for a broad scientific audience.

- **Referee #2:**

We thank the authors for addressing our previous comments and for the improvements made to the manuscript. The revised figures, particularly the separation of distinct mitochondrial dysfunctions, enhance clarity and allow for more focused interpretation.

To further strengthen the visual impact and utility of the figures, we recommend indicating where key therapeutic strategies are proposed to act. This would provide direct insight into the mechanisms of intervention and move the figures beyond a list-like summary toward more integrative illustrations. For example:

- In Figure 2, it would be helpful to indicate the points of action for Coenzyme A and NAD⁺ precursors.
- In Figure 3, including urolithin A with an arrow pointing toward PINK1 would clarify its role in modulating mitophagy.
- In Figure 4, we suggest briefly explaining the terms "proximal" and "distal" in the figure legend, in addition to the main text, to support reader understanding.
- In Figure 5, we recommend reintroducing the microvesicles injection with mitochondria as in the previous manuscript.

These additions would help highlight relevant therapeutic avenues.

Reply to reviewer's point: We sincerely thank the reviewer for their insightful suggestion regarding the integration of therapeutic strategies into the figure. We fully agree that this addition enhances the clarity and impact of the review. In response, we have incorporated the relevant therapeutic strategies into the individual figures. Specifically, we have updated the figure legend for Figure 4 as follows: 'Figure 4. Asymmetric Inheritance of Mitochondria and its therapeutic potentials. Following activation by antigen-presenting cells (APCs), T cells form polarity and undergo mitosis with a proximal and distal side to APCs. During this process, mitochondria can be inherited through asymmetric cell division (ACD). Selective mitochondrial inheritance through ACD offers a potential mechanism for clearing damaged mitochondria. Transient rapamycin treatment

enhances ACD efficiency. Improving ACD also holds promise for improving CAR-T cell therapy and vaccination response. The figure was created with Biorender.com.

We also attached the revised figures(Figure 2, 3, 5) as below:

• **Mitochondrial degradation by mitophagy**

• **Therapeutic Strategies**

Inducing mitophagy: Urolithin A, Ceramide.

Potential mitophagy inducers: spermidine, USP30 inhibitors, NRF2 inducers, metformin, AMPK inducers, NMN.

Other strategies:

Lifestyle: Caloric restriction or long-term exercise.

• **Referee #3:**

The authors have adequately addressed my comments and the review can now be published.

Reply to reviewer's point: We thank the reviewer for the comments and efforts to make this review better. We also appreciate the support in allowing us to move forward with the publication.

khia

Nuffield Department of Orthopaedics,
Rheumatology and Musculoskeletal Sciences

Professor AK Simon, PhD

Anna Katharina Simon
University of Oxford
The Kennedy Institute of Rheumatology, NDORMS
United Kingdom

Dear Dr. Simon,

I am pleased to inform you that your review has been accepted for publication in EMBO reports. Your manuscript will be processed for publication by EMBO Press. It will be copy edited and you will receive page proofs prior to publication.

You will soon be contacted by Springer Nature to sign your publishing license. When you login to the customer service website, please use the token/code copied below to waive the article publication charges. Should you experience any difficulty, please email publishing@embo.org.

XXXXXXXXXXXXXXXXXXXXXXXXXXXX

Yours sincerely,
